# Evaluation of snow data assimilation using the Ensemble Kalman Filter for seasonal streamflow prediction in the Western United States

Chengcheng Huang[1, 2], Andrew J. Newman[2], Martyn P. Clark[2], Andrew W. Wood[2]

and Xiaogu Zheng[1]

[1] College of Global Change and Earth System Science, Beijing Normal University, Beijing, China

[2] National Center for Atmospheric Research, Boulder CO, 80301, USA

*Correspondence to*: Andrew J. Newman (anewman@ucar.edu)

**Abstract.** In this study we examine the potential of snow water equivalent data assimilation (DA) using the ensemble Kalman Filter (EnKF) to improve seasonal streamflow predictions. There are several goals of this study. First, we aim to examine some empirical aspects of the EnKF, namely the observational uncertainty estimates and the observation transformation operator. Second, we use a newly created ensemble forcing dataset to develop ensemble model states that provide an estimate of model state uncertainty. Third, we examine the impact of varying the observation and model state uncertainty on forecast skill. We use basins from the Pacific Northwest, Rocky Mountains, and California in the western United States with the coupled Snow17 and Sacramento Soil Moisture Accounting (SAC-SMA) models. We find that most EnKF implementation variations result in improved streamflow prediction, but the methodological choices in the examined components impact predictive performance in a non-uniform way across the basins. Finally, basins with relatively higher calibrated model performance (> 0.80 NSE) without DA generally have lesser improvement with DA, while basins with poorer historical model performance show greater improvements.

*Keywords:*

Hydrological data assimilation; SWE; EnKF; Snow-17; SAC

## 1   Introduction

In the snow-dominated watersheds of the Western US, spring snowmelt is a major source of runoff (Barnett et al., 2005; Clark and Hay, 2004; Singh and Kumar, 1997; Slater and Clark, 2006). In such basins, the initial conditions of the basin, primarily in the form of snow water equivalent (SWE), drive predictability out to seasonal time scales (Wood et al., 2005; Wood and Lettenmaier, 2008; Mahanama et al. 2012; Staudinger and Seibert 2014; Wood et al. 2015). Thus better estimates of basin mean initial SWE should lead to better seasonal streamflow predictions (Arheimer et al., 2011; Clark and Hay, 2004; Slater and Clark, 2006; Wood et al. 2015). For various reasons (e.g., the uncertainty in model parameters, forcing data, model structures), simulated SWE in hydrological models can be very different from reality (Pan et al., 2003). Fortunately, a variety of snow observations (including point gauge and spatial satellite data) contain valuable information (Andreadis and Lettenmaier, 2006; Barrett, 2003; Engeset et al., 2003; Mitchell et al., 2004; Su et al., 2010; Sun et al., 2004).

Many studies have explored the role of snow data assimilation in different modeling frameworks (Moradkhani, 2008; Takala et al., 2011; McGuire et al, 2006; Wood and Lettenmaier, 2006). Of particular focus here are papers that have examined the impact of SWE data assimilation (DA) on runoff modelling and prediction (e.g. Bergeron et al., 2016; Griessinger et al., 2016; Wood and Lettenmaier, 2006; Franz et al., 2014; Jörg-Hess et al., 2015; Moradkhani, 2008; Slater and Clark, 2006). Among the major challenges facing SWE-based DA are that the time-space resolution of remote sensing SWE data are too coarse or period-limited for many watershed-scale hydrological applications in mountainous regions (Dietz et al., 2012; Jörg-Hess et al., 2015), and point gauge snow data have sparse and uneven spatial coverage (Slater and Clark 2006). For point measurements, spatial interpolation of SWE measurements is typically used to estimate observed SWE state in a watershed of interest (e.g.,Franz et al., 2014; Jörg-Hess et al., 2015; Slater and Clark, 2006; Wood and Lettenmaier, 2006).

Here we use the Ensemble Kalman Filter (EnKF) method for DA using an implementation that allowing for seasonally varying estimates of observation and model error variances (Evensen, 1994, 2003; Evensen et al., 2007). The EnKF framework has been successfully implemented in research basins in several previous studies (Clark et al., 2008; Franz et al., 2014; Moradkhani et al., 2005; Slater and Clark, 2006; Vrugt et al., 2006). The EnKF provides an objective analytical framework to optimize the update of model states based on observed values and their corresponding uncertainties. While the EnKF approach has a formal theory, its overall objectivity in an application (contrasting with an arbitrary DA approach such as direct insertion) nonetheless depends on several methodological choices that are often empirical when applied to SWE DA.

Following Slater and Clark (2006), this study uses two slightly different approaches to estimate ensemble SWE observations with point gauge SWE data from surrounding gauge sites for study basins. When using calibrated hydrologic modeling systems, model SWE states may exhibit systematic biases from observed SWE estimates for a number of reasons – e.g., all hydrologic models must simplify real watershed physics and structure, and model parameter estimation (calibration) may result in SWE

behavior that in part compensates for forcing or model errors (e.g. Slater and Clark, 2006). Therefore, transformation of snow observations to model space is needed before they are used to update the model states to ensure that the model ingests SWE estimates that are as close to unbiased relative to the model climatology as possible. We explore two variations on an approach using cumulative density function (CDF) transformations of observations to model space (following Wood and Lettenmaier, 2006, among others). Additionally, we undertake a sensitivity analysis to highlight the importance of robust observations and model uncertainty estimates. We focus on the impacts of updates made just once per snow accumulation season, noting that an important choice that is not examined as a result is the selection of DA dates and frequency. For a given generally optimal selection of the EnKF approach, the Ensemble Streamflow Prediction (ESP) approach is used to test the impact of SWE DA on subsequent streamflow forecasts.

For context, operational seasonal streamflow forecasts in the US currently do not use formalized DA. If the initial states of the model are suspected to contain error (He et al. 2012), DA is performed through subjective forecaster intervention. Manual adjustments (termed 'MODs', e.g. Anderson 2002) to model states (e.g. SWE) are applied repeatedly throughout the water year, and particularly before initializing seasonal forecasts. This manual nature of the correction hinders the ability to scale up DA procedures to many basins, to benchmark DA performance, and quantify improvements to the forecast system as skill depends on the forecaster's experience (Seo et al. 2003).

The central motivating aim of this study is thus to assess the potential benefits of objective, automated SWE DA against a reference model configuration to identify forecast improvement opportunities. We apply the EnKF DA approach to nine river basins in the Western US that have a range of basin features and environmental conditions, over a period of multiple decades. This experimental scope differs from many previous studies that focus on one or two basins (e.g., Clark et al., 2008; Franz et al., 2014; He et al., 2012; Moradkhani et al., 2005), or assess DA performance over shorter periods. We also use ensemble simulations driven by a new probabilistic forcing dataset (Newman et al, 2015) as a basis for estimating model SWE uncertainty, in contrast to prior studies that relied on more arbitrary distributional assumptions. This range of basins permits us to explore the question of: "In what types of basins might automated SWE DA improve seasonal streamflow forecasts?"

Additionally, as discussed throughout the introduction, the EnKF approach has several empirical components that require tuning. We therefore examine performance sensitivities related to three elements: 1) the estimation of watershed mean SWE from surrounding point measurements; 2) the transformation operator that relates watershed mean SWE to model mean SWE; and 3) sensitivity analyses of the relative size of observed and model error variance.

The following sections discuss the study basins and data sets, and the model and EnKF DA approach, before the presenting study results and discussion, and a summary.

## 2 Study basins and data

In this study, nine basins across the Western US are selected for SWE DA evaluation. They are in the Pacific Northwest, California (Sierra Nevada Mountains), and central Rocky Mountains. We focus on these three areas as they span a range of snow accumulation and melt conditions of the Western US and are in areas with active seasonal streamflow prediction and water resource management. We do not examine rain driven low-lying basins because they do not have significant SWE contributions to runoff. The locations of the basins and nearby SWE gauge sites are shown in Figure 1, illustrating that all of the study watersheds have SWE measurements distributed in and/or around the basins. The main features of these basins are shown in Table 1. The basin areas range from 16 to 1163 km$^2$ and the mean elevations of the basins range from 998 to 3459 m with a large spread in basin mean slopes (as estimated from a fine-resolution digital elevation model) and forest percentage. Two sources of SWE observations are used in this study: (1) the widely used Snow Telemetry (SNOTEL) network for Natural Resources Conservation Service (NRCS), which covers most of the western US; and (2) the California Department of Water resources (DWR, denoted as CADWR sites hereafter), which maintains a snow pillow network for California. The SWE data from CADWR sites have frequent missing data and some unrealistic extreme values, thus extensive manual quality control was required before using the CADWR data in the study.

## 3 Methodology

### 3.1 Models and calibration

The Snow-17 temperature index snow model is coupled to the Sacramento Soil Moisture Accounting (SAC-SMA) conceptual hydrologic model (Anderson, 2002; Anderson, 1973; Burnash and Singh, 1995; Burnash et al., 1973; Franz et al., 2014; Newman et al., 2015a) to simulate streamflow in this study. This model combination has been in operational use by US National Weather Service (NWS) River Forecast Centers (RFCs) since the 1970s (Anderson, 1972; 1973). The Snow-17 model is a conceptual snow pack model that employs an air temperature index to partition precipitation into rain and snow and parameterize energy exchange and snowpack evolution processes. The only required forcing inputs are near-surface air temperature and precipitation. The output rain-plus-snowmelt (RAIM) time series from Snow-17 is part of the forcing input of the SAC-SMA model. SAC-SMA is a conceptual hydrologic model that uses five moisture zones to describe the movement of water through watersheds. The required forcing input is the potential evaporation and the surface water input from Snow-17.

Daily streamflow data from United States Geological Survey (USGS) National Water Information System server (http://waterdata.usgs.gov/usa/nwis/sw) are used to calibrate 20 parameters of Snow-17 and SAC-SMA model. The calibration is obtained using the shuffled complex evolution global search algorithm (SCE; Duan et al, 1992) via minimizing daily

simulation Root Mean Square Error (RMSE). USGS streamflow data are also used to verify the model predictions.
Model uncertainty arises from model parameter and structural uncertainty (e.g. Clark et al., 2008) and forcing input uncertainty
(e.g., Carpenter and Georgakakos, 2004). Focusing on the latter, we drive the hydrology models with 100 equally likely
members of meteorological data ensemble generated as described in Newman et al. (2015b), producing an 100 member
ensemble of model moisture states, including SWE, and streamflow. The daily-varying spread of the ensemble model states
serve as the estimate of model uncertainty. Because this method estimates SWE uncertainty without also considering sources
other than forcing input uncertainty, and therefore may underestimate model uncertainty in initial SWE (e.g. Franz et al. 2014),
we also include a sensitivity analysis to explore the sensitivity of DA results to variations in the estimated observation and
model uncertainty magnitudes.
**3.2 Generating ensembles of estimated observed watershed SWE**
Since the SWE gauge observations are point measurements that do not represent the watershed mean conditions and have
observation error, observation uncertainty needs to be robustly estimated to ensure reasonable DA performance. In this study,
we follow Slater and Clark (2006) to generate ensemble estimated catchment SWE from gauge observations using a multiple
linear regression in which the predictors are the attributes of SWE gauge sites (longitude, latitude and elevation). The
observation uncertainty is estimated by leave-one-out (LOO) cross validation: i.e., each station is left out of the regression
training and then its SWE is predicted and verified against its actual measurement. For reducing interpolation uncertainty
caused by spatial heterogeneity of SWE gauge sites, the SWE values are transformed into percentiles or Z-scores (eg, standard
normal deviates) before the regression is performed, and the corresponding inverse transformations are used to convert them
back to SWE values. These two approaches are denoted as percentile and Z-score interpolation respectively and detailed
descriptions for them are as follows.
**3.2.1 Percentile interpolation**
First, the non-exceedance percentile $p_y^o(k)$ of each SWE observation (observation based values noted with superscript o) at
gauge site $k$ on DA date in year $y$ is calculated based on its rank, or percentile, within a sample of all SWE observations in all
years at the same site within a time-window of +/- n days centered on the date of the observation in each year.
Then we use the percentiles to do linear regression on geographic features latitude, longitude and elevation to estimate the
SWE percentile for the target basin: $\hat{p}_y^o$, where the hat indicates the basin mean estimate. By LOO cross validation, the
interpolation error of the linear regression is estimated as $\hat{e}_y^o$. We sample from normal distribution $N(\hat{p}_y^o, \hat{e}_y^o)$ to get the
ensemble percentiles $\{\hat{p}_y^o(j)\}$, where $j = 1,\ldots, 100$ represents ensemble member.
Finally, we take the corresponding $\hat{p}_y^o(j)$ percentile from the full ensemble model SWE within the time-window of +/- n
days centered on the DA date each year in all years, denoted as $\hat{S}_y^f(j)$. The final ensemble SWE observations on DA date at
year $y$ for the target basin are $\{\hat{S}_y^f(j)\}$, where $j = 1,\ldots,$ 100.
**3.2.2 *Z*-score interpolation**
First, we use the observed SWE at gauge site $k$ on DA date in year $y$ to calculate the *Z*-score:
$$Zscore_y(k) = \frac{S_y^o(k) - \overline{S^o(k)}}{\sigma(S^o(k))},\tag{1}$$
where $\overline{S^o(k)}$ and $\sigma(S^o(k))$ are the long-term mean and standard deviation of a sample of all non-zero SWE observations at
the same site within a time-window of +/- n days centered on the date of the observation respectively. Here we use the *Z*-score
in the linear regression and again use LOO cross validation to estimate the mean and interpolation error of the *Z*-score for a
target basin. Then we sample from normal distribution to get ensemble *Z*-scores for target basin, denoted as $\{\hat{Z}\text{-}score_y^o(j)\}$,
where $j = 1,\ldots,$ 100 represents ensemble member. Finally we use the following equation to transform Z-score to back to SWE
values:
$$\hat{S}_y^o(j) = \hat{Z}score_y^o \times \sigma(S^f(k)) + \overline{S^f(k)},\tag{2}$$
where $\overline{S^f(k)}$ and $\sigma(S^f(k))$ are the long-term non-zero mean and standard deviation of the full ensemble model SWE within
the time-window of +/- n days centered on the DA date each year in all years respectively. The final ensemble SWE
observations on DA date at year $y$ for the target basin are $\{\hat{S}_y^o(j)\}$, where $j = 1,\ldots,$ 100.
Both percentile and Z-score transformations normalize the original SWE values to decrease their spatial variability (Slater and
Clark 2006; Wood and Lettenmaier, 2006). The latter ensures the ensemble observations have the same mean as the ensemble
model SWE and the variance of ensemble observations is proportional to ensemble model SWE variance. The former
emphasizes the shape of the observation time series. SWE observations in and near a watershed but at different elevations may
have greatly varying values, but their percentile and Z-score statistics will show reduced variation because they arise from
similar relative weather conditions with respect to conditions in other years. Using normalized statistics significantly reduces
the interpolation uncertainty and systematic biases relative to the watershed's SWE climatology.
**3.3 EnKF approach and experimental design**
For evaluating the relative performance of DA and for re-initializing the soil moisture of DA runs at the beginning of each
water year (WY), an open loop or 'control' retrospective simulation (denoted No DA) is performed using the calibrated model
parameters with ensemble forcing data. This control run is one continuous simulation per ensemble member for the entire
hindcasting and evaluation period (1981-201X) for each basin. Because this study focuses on assessing variations in
methodological aspects of the DA approach rather than differences in performance throughout a forecasting season, we apply

DA updates only once per year, using the date on which the SWE correlation with future runoff is highest for the study basin, but no later than 1 April, a common date for initiation of spring seasonal runoff forecasts.

The EnKF method used in this study is a time-discrete forecast and linear observation system described by two relationships (generally following the notation of Ide et al. (1997) and Wu et al. (2012)) :

$$x_{i+1}^t = M(x_i^t) + \eta_i, \tag{3}$$

$$y_i^o = h(x_i^t) + \varepsilon_i, \tag{4}$$

where $i$ is the time step, M is the coupled Snow17 and SAC-SMA model, $x$ is the state variable and $y$ is the observation variable (in this study both $x$ and $y$ are the one-dimensional vector containing basin mean SWE for the target watershed across all ensemble members), the superscripts $t$ and $o$ stand for truth and observed respectively, $\eta$ and $\varepsilon$ are the model and observation errors respectively, and $h$ is the observation operator that maps the model states to the observation variable. In this study, $h$ is simply the identity vector as we regard the SWE estimates that have been transformed to model space as observation $y$, as a pre-processing step.

The SWE DA approach is implemented via the following procedure:

    1)  Run the watershed model once for each ensemble forcing member from the beginning of a WY until the DA date with initial states $x_0$ taken from the retrospective control runs, producing the ensemble forecast states $x_i^f$. The superscript $f$ denotes forecast.

    2)  Calculate the ensemble analysis states:

$$x_i^a = x_i^f + s_i h_i^T (h_i s_i h_i^T + o_i)^{-1} d_i, \tag{5}$$

where superscript $a$ means analysis, $o$ and $s$ are the observed and model simulation error variances (estimated by the variance of ensemble observations and model states respectively) respectively, and the innovation vector (residual) is calculated as:

$$d_i = y_i^o - h_i(x_i^f), \tag{6}$$

    3)  Update the Snow-17 SWE states with the analysis states to use for initialization of forecasts through the end of the WY.

Steps 1-3 are repeated for all WY available in the hindcast period (1981-201X). Soil states are re-initialized using the states from the retrospective (No DA) run at the start of every WY (October 1), when there is no SWE. To summarize, we calculate an analysis via Eq. 5 and use that analysis to update the Snow-17 SWE states. We then run the model with the updated states until the end of the WY.

**3.4 Model and observation error variance**

In this study, only the uncertainty of the forcing data is taken into account in our model uncertainty, and uncertainty that arises from model structural and parameter errors could cause the true model error to be larger. Thus we assess the impacts of inflating model error variance to evaluate the relative size of observed and forecast error variance. We simply set the model SWE error

variance to 1/2 and 2 times of the original size to see how the DA performances change. If increasing the model error variance results in DA performance improvements, it would indicate that the model error variance is underestimated, and vice versa. This sensitivity analysis underscores the importance of a careful effort to properly estimate both model and observational uncertainty when using the EnKF – a challenge that is well known in the DA community.

**3.5 Seasonal Ensemble Streamflow Prediction**

Although the impacts of the SWE DA on forecast accuracy can be assessed through verification of post-adjustment simulations using 'perfect' future forcing, we demonstrate the performance of SWE DA by initializing seasonal ESP forecasts for a streamflow forecast product that is widely used in water management, the snowmelt-period runoff volume from April through July. ESP uses historical climate data to represent the future climate conditions each year from the start point of forecast period to predict streamflow. Two typical ESP applications are tested in this study. Because we have an ensemble of historical forcing instead of the traditional application in which only a single historical forcing time series is available, there are different ways to construct an ESP. We adopt two: (1) We construct the ESP forcing ensemble by randomly selecting one year of the historical ensemble forcing data for each historical member of the ESP; and (2) We use all historical years of ensemble mean forcing data for each ESP historical year member, yielding a 30*100 member ensemble for an ESP based on meteorology from 1981-2010 (variations are noted ens forcing and ens mean forcing respectively in subsequent figures discussing ESP results).

**3.6 Verification metrics**

In this study, five frequently used statistics are calculated for April through July seasonal streamflow volume expressed as runoff (mm) for evaluating the two DA approaches. The bias, correlation coefficient (R), relative root mean squared error (R-RMSE), Nash-Sutcliffe efficiency (NSE) are based on the ensemble averages. The continuous ranked probability score (CRPS) is a measurement of error for probabilistic prediction (Murphy and Winkler, 1987). It is defined as the integrated squared difference between cumulative distribution function (CDF) of forecasts and observations:

$$\text{CRPS} = \int_{-\infty}^{+\infty} \left[ F^{\text{f}}(x) - F^{\text{o}}(x) \right]^2 dx, \tag{7}$$

where $F^{\text{f}}$ and $F^{\text{o}}$ are CDFs for forecasts and observations of streamflow respectively. Small CRPS values mean more accurate forecasts, with 0 value indicating a perfect forecast accuracy.

**4 Results and Discussion**

**4.1 Overall performance in the case basins**

Using the two approaches described in Section 3.2 with three different window lengths (7 days, 3 months, 1 year), a sample comparison from one year (2004) of the results for estimated watershed SWE from the two methods versus the model SWE ensemble on DA date (DA dates for the case basins are listed in Table 1) for the case basins are shown in Figure 2. The distributions of SWE from the model ensemble and from the percentile and *Z*-score interpolation methods differ in ways that

are not consistent across all watersheds. The variance of the estimated observed SWE for both methods is generally largest for the 1-year, an effect that is more pronounced for the *Z*-score interpolation. However, we also note that the ensemble observations of 7-day window can have a larger variance than the 3-month window, and as large as the 1-year window in some cases. See the percentile interpolation for the Payette River for 7-d window in Figure 2 where the 7-day window interquartile range is about 250 mm, the 1-year window range is 300 mm while the 3-month window is only about 120 mm. This is likely due to the more limited sample size for the regression, which can reduce the positive impact of DA performance. For example, the SF Payette River and the Greys River have positive DA impact for both the 7-day and 3-month windows but for the 7-day window the positive impact is reduced by roughly half in both basins for most metrics (Tables S.1 and S.3 of Supplement S1). Increased estimated observation variance decreases the weight of the observations in an EnKF approach and thus decreases the impact of the observations. In this study, a 3-month window of SWE observations generally gives the best performance. However, in some basins a different window length may bring larger improvements. Longer windows mean that the transformation is more statistically representative of the long-term model-observation climatology. Shorter time windows imply that the model SWE values used for transformation are more relevant to a specific seasonal time period, avoiding aliasing for seasonality, but have much smaller sample sizes and may not properly represent the relationship between model and observation climatologies. The window length must be a balance between these two considerations. Therefore, a 3-month window is recommended for both approaches.

The evaluation statistics for simulated streamflow using perfect forcing after DA with ensemble SWE observations estimated by the percentile and Z-score interpolation approaches for the 3-month window are shown in Figures 3 and 4. They are also compiled in Tables S.1-6 in supplement S1. In those tables, the $2^{nd}$ column shows the forecast error variance used to calculate analysis states, where "No DA" means the open loop control run (see Section 3.3), and the P, 1/2·P and 2·P refer to the DA runs with the model error variance estimated by 1, 1/2 and 2 times the original size of the ensemble model variance. Both percentile and *Z*-score interpolation approaches exhibit enhanced DA performance among the case basins, indicating that both approaches are effective in adding observation based information to the model simulations. Overall, using the original model variance estimate (case P) the mean improvement for the percentile interpolation method (Z-score method) is a reduction in relative RMSE (R-RMSE) of about 11% (12%) and an increase in NSE of 0.03 (0.05). The percentile interpolation and Z-score interpolation methods vary in performance across the basins with both performing better in some basins and not others (e.g, percentile interpolation performs slightly better than Z-score interpolation in Grey River using NSE as the evaluation metric (0.94 vs 0.93) and slightly worse that in SF Tolt River (0.82 vs 0.88)). Using NSE, percentile interpolation performs better in the Greys River, while Z-score interpolation performs better in the Vallecito, South Fork of the Tolt, Merced, and Smith Rivers. To the hundredth NSE value (0.01) both methods are equivalent in the South Fork of the Payette River, and

General and Blackwood Creeks.
The results of forecast error variance inflation shows that for both percentile and Z-score interpolation, "2·P" has better
performance than "P" in most of the case basins – i.e., increasing the model error variance leads the assimilation to trust
observations more and improves the DA performance (circles in both figures generally have improved evaluation metrics than
squares or triangles). Using NSE, the percentile (Z-score) interpolation "2·P" case is on average another 0.01 (0.01) better than
the "P" case across the nine basins. This sensitivity analysis of model uncertainty impacts on DA performance suggest that
either the forcing-alone based estimation of model errors underestimate the total model error variance, or the observed SWE
error estimation approaches (interpolation plus the SWE regression) tend to overestimate observation uncertainty, or both. It
is likely we are underestimating model uncertainty because we have not taken model structural and parameter uncertainty into
consideration. Both approaches bring incremental enhancements to the ensemble mean streamflow hindcast in most basins
when evaluated across the R-RMSE, R and NSE metrics, however DA does not help correct forecast biases in these simulations.
Post-processing procedures (e.g. bias correction) could be used to further enhance the forecast performance, but is not a focus
of this study. These figures also show that forecasts without DA ("No DA" in figures, "NoDA" in text) that have relatively
better performance, mostly due to better simulations of forecast initial conditions, benefit less from DA. Three of the basins
have a NoDA seasonal runoff NSE of less than 0.8, with an average improvement of 0.05 for the percentile regression and
0.12 for the Z-score regression versus 0.03 and 0.05 across all nine basins. Four basins have seasonal runoff NSE values of at
least 0.89 and the two DA methods result in minimal improvement, 0.02 for both methods. With a sample size of nine, little
statistical significance can be attached to these results, but they do suggest DA is more beneficial in poorly calibrated basins.
Future work will examine the potential for DA based on NoDA (open loop) model performances and the characteristics of
nearby observed SWE data.
Figure 5 summarizes the ESP evaluation statistics. For simplicity, only the percentile interpolation approach with a 3-month
window is shown without forecast error inflation. It shows that for both ESP forcing methodologies used (Section 3.5) in all
the case study watersheds, SWE DA enhances seasonal runoff prediction skill, including the probabilistic prediction metric
CRPS. Again, higher skill NoDA watersheds saw smaller DA improvements. The DA evaluation metric improvement
increment versus the corresponding NoDA evaluation metric score for the case basins are shown in Figure 6. The DA
improvements in all evaluation metrics have a generally weak negative correlation with NoDA performance, which again
highlights that better simulated basins benefit less from SWE DA.
**4.1.1 Broader DA Potential**
In general, the incremental DA improvements are relatively smaller where the NoDA model performance is relatively better.
However, specific basin performance is dependent on many factors including: 1) representativeness of nearby observations to

basin conditions; 2) quality of observations; 3) specific basin characteristics of the calibrated hydrologic model. Because we use calibrated, watershed scale hydrologic models, transferability of performance characteristics of the DA approach without implementation in each basin is limited. That being said, Figure 7 displays the difference between the rank correlation of SWE and runoff for the calibrated model (NoDA) and highest correlated observation site (from the nearest 10 sites). It highlights the same general spatial patterns seen in the 9 basins simulated here. The potential for larger DA improvement appears to be in the Pacific Northwest (upper left of figure). Basins in the Dakotas (upper right basins) are far from SNOTEL sites and have little areal SWE; basins along the far southern US have little SWE and runoff as well. Throughout the central Rockies (central basins), model-observation correlation differences are small, potentially indicating reduced DA improvement potential, in agreement with the results seen above.

**4.2 Case study analyses**

To provide a more in-depth examination of the SWE DA impacts to the watershed model states and fluxes, time series of runoff and SWE are shown in Figures 8, 9 and 10 for three example basins, one for each region (the same figures for the other six basins are included in the supplemental material), and for one hindcast year. The feedback from the change of SWE on DA date to seasonal runoff is readily apparent. Increasing the ensemble model SWE through DA will lead to increased model runoff, and vice versa. For basins with a strong seasonal cycle of streamflow (e.g. Greys and Merced River), SWE DA may improve daily runoff forecasts in years when seasonal volume forecast improvements are seen, although this is not true in every watershed (e.g. Tolt River). For example, the daily NSE for the Greys River in 1997 after DA was improved from 0.53 to 0.80 in the perfect forcing example, and this is via bias reduction as the daily flow time series is unchanged. In Figure 9, the NSE of the daily flow prediction of the Tolt River is essentially unchanged (0.54 for DA, 0.53 for NoDA) even though the seasonal volume prediction is improved (1990 mm observed, 1968 mm DA, 1534 mm NoDA). In this case improvements to bias did not improve NSE as the bias improvements did not improve the squared daily flow differences (e.g. RMSE: 7.76 vs 7.88 for DA vs NoDA).

Figures 11, 12 and 13 show several scatter plots of forecast period runoff for the ESP ensemble forcing and perfect forcing forecasts, versus observed runoff, in the three case basins for all of the hindcast years. The left two columns show the comparison for NoDA and DA simulated seasonal runoff vs observed runoff for perfect (top row) and ESP ensemble forcing (bottom row) respectively. The 1:1 lines are shown as grey dashed lines and regression lines for the results are shown as green solid line. The results after DA have higher correlation and are generally closer to the 1:1 line, which indicates that for both forcing types SWE DA improves seasonal runoff simulation and prediction skill. The rightmost columns in these three figures show the scatter plots of SWE increment (i.e., SWE analyses states minus model SWE without DA) vs runoff error (i.e., the simulated seasonal runoff without DA minus the observed seasonal runoff). If the runoff errors are positive (the seasonal runoff

is overestimated), we would expect the SWE increment to be negative in order to decrease the model seasonal runoff
(counteract model error) and vice versa. Thus the ideal results are that the points fall onto different sides of $y$=0 and $x$=0 lines
(shown as grey dashed lines in this panel), i.e., the points all fall into the 2nd (upper left) and 4th (lower right) quadrants. This
is generally the case for our case basins for both perfect and ESP forcing, which again shows that the SWE DA approach is
successful in reducing model and forecast error.
For the three basins highlighted here, there are years where the DA SWE increment is not in the 2nd or 4th quadrants. In these
years, the increment decreases subsequent forecast skill. Overall, there are 11 of 28 (39%), 4 of 24 (17%), and 12 of 26 (46%)
years for the Greys, Tolt and Merced rivers where this is the case using perfect forcing. These years generally correspond to
small SWE increments relative to that year's SWE and runoff in all basins except for five years in the Merced River where the
SWE increment is larger than 10% of that year's streamflow production and incorrect. In the Greys River, all incorrect
increments are less than 10% of the observed runoff for that year and also in years where the NoDA runoff error is less than
10% of observed. A small increment implies that the estimated observed and model SWE are very similar, and thus in years
with small model error, the model SWE climatology closely matches observed climatology after transformation for this basin.
Figure 14 highlights an example WY in the Merced River where the SWE increment and runoff error are both negative,
indicating that DA increased the model forecast error.
The Merced River is the only basin to use state of California SWE observations, and these may be of lower quality as evidenced
by the large amount of manual quality control we had to perform on the data and the discussion of these data in Lundquist et
al. (2015). This suggests that observed SWE data need to be of higher quality (or information content) than the calibrated
model SWE to have the positive impact in the DA approach. The calibrated Merced model has -19% April-July runoff bias
with 23 (88%) of years having a negative runoff error. EnKF SWE increments are negative in 15 (58%) and positive in 11
(42%) of the years. This indicates that the observed SWE transformation to model space is largely unbiased, but the calibrated
model bias impacts SWE DA performance. Calibration of the model specifically for seasonal flow to ensure minimal bias, or
hydrologic parameter estimation within the EnKF approach (e.g. He et al. 2012) would likely improve hydrologic model
performance and thus seasonal SWE DA forecasts in the Merced. Finally, examination of El Nino/La Nina signals (not shown)
revealed no clear pattern with degradation of DA forecast skill.
Finally, there are years where the NoDA runoff error is large, but the SWE increment is small in all three basins. This is not
unexpected as spring SWE is not perfectly correlated with subsequent runoff. This may also hint at a level of data loss in the
EnKF approach, and future work should compare streamflow hindcasts using this type of DA approach with traditional
statistical methods using SWE as a primary input. It also suggests that improved model calibration, or in combination with
model parameter estimation in the EnKF approach (e.g. He et al. 2012) may improve DA performance across all basins, not
just the Merced.
**5 Summary and Conclusions**
This study tests variants of EnKF SWE DA approaches in 9 case basins in Western US. These basins have seasonal runoff
representative of basins used for water resource management across the Western US and have at least 6 close SWE gauge sites
with 20+ years of observation history. Two approaches of constructing SWE ensemble observations, percentile and Z-score
interpolation, are examined in this study in an effort to reduce the spatial variability and decrease the interpolation uncertainty
while also transforming the observations to model space (e.g., the range of the model climatology). A 3-month window of
SWE observations generally gives the best performance for these two approaches in this study (Figs. 2-4, Tables S.1-6 in S1).
However, in some basins a different window length may bring larger improvements. A suitable window length needs to include
sufficient samples for transformation as well as including the most relevant samples (i.e., a specific seasonal time period).
Sensitivity analyses of model uncertainty impacts on DA performance suggest that either the forcing-alone based estimation
of model errors underestimate the total model error variance, or the observed SWE error estimation approaches (interpolation
plus the SWE regression) tend to overestimate observation uncertainty, or both (Figs. 3-4, Tables S.1-6 in S1) . Future work
should examine this in more detail, as this work clearly indicates that uncertainty scaling approaches (for the model and/or the
observations) are likely to be a valuable step for further DA improvements.
Encouragingly, the ESP-based assessment of automated SWE DA in the case study watersheds shows clearly the potential for
SWE DA to enhance seasonal runoff forecasts, which is notable as the objective incorporation of observed SWE has been a
long-standing challenge in operational forecasting. We show at least minor improvement in seasonal runoff forecasts in all
nine basins (Figs. 5-6). A notable finding is also that the benefits of SWE are linked to the quality of the model simulations of
the basin, which can help to target the application of DA to locations where it will have the most benefit (Figs. 5-6). For the
basins with poor no DA simulations (e.g., the SF Tolt River Fig. 12), the SWE DA can potentially have greater model
performance impacts. The Pacific Northwest and California was found to have the greatest potential for DA improvements to
seasonal forecasting in this study (Fig. 7). This stems from weaker NoDA model performance; the NoDA model run will have
more years with larger runoff errors.  However, there are still individual years where DA may not improve the forecast.  This
likely stems from hydrologic model bias that leads to SWE state corrections enhancing rather than reducing runoff errors (e.g.
Merced River, Figs. 13-14).
We chose a DA update frequency of once per year, the date of climatological maximum correlation of modeled and observed
runoff.  In operational practice, updates would be applied more frequently, pointing to an area for future research. We note also
that this study was conducted using conceptual lumped watershed models, similar to those used in operational practice in the
US. As a result, this study does not shed light on how to address additional challenges that may be associated with using SWE

DA in spatially distributed models, or with spatially continuous datasets (e.g., satellite and remote sensing SWE estimates) that are increasingly being developed or applied in streamflow forecasting contexts. SWE DA has been implemented in distributed models in prior experimental contexts across large domains (e.g., Wood and Lettenmaier, 2006), but a systematic examination of EnKF DA in spatially distributed hydrological models, coupled with a thoughtful accounting for model parameter and structural errors remains a potentially fruitful area of research and development.

**Data Availability**

All data used in this study are publicly available. The watershed shapefiles and basin information are described in Newman et al. (2015a) at: doi:10.5065/D6MW2F4D. The forcing ensemble is described in Newman et al. (2015b) and are available at: doi:10.1065/D6TH8JR2. The streamflow data are available through the USGS via: http://waterdata.usgs.gov/usa/nwis/sw and in doi:10.5065/D6MW2F4D. The SNOTEL observations are available at: www.wcc.nrcs.usda.gov/snow/ while the California SWE observations are available at: cdec.water.ca.gov/snow.

**Acknowledgements**

This work was supported by China Scholarship Council (No. 201406040164), the NCAR/Research Applications Laboratory, the US Department of the Interior Bureau of Reclamation, and the US Army Corps of Engineers Climate Preparedness and Resilience Program.

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

**Table 1** Basin features of nine case basins.

| Region | Basin ID | Elevation (m) | Minimum elevation (m) | Maximum elevation (m) | DA date | Basin area (km²) | Slope (m km⁻¹) | Forest fraction | Basin name |
|---|---|---|---|---|---|---|---|---|---|
| 14 | 09081600 | 3092.15 | 2050 | 4250 | April 1 | 436.88 | 150.58 | 0.61 | Crystal River |
| 14 | 09352900 | 3459.15 | 2450 | 4250 | April 1 | 187.74 | 156.09 | 0.52 | Vallecito Creek |
| 17 | 13023000 | 2468.57 | 1750 | 3450 | March 1 | 1163.72 | 98.51 | 0.68 | Greys River |
| 17 | 12147600 | 998.25 | 550 | 1650 | April 1 | 16.07 | 159.37 | 1 | SF Tolt River |
| 17 | 13235000 | 2077.16 | 1150 | 3250 | April 1 | 1158.47 | 126.25 | 0.86 | SF Payette River |
| 17 | 14158790 | 1210.48 | 750 | 1750 | March 15 | 40.76 | 116.44 | 1 | Smith River |
| 16 | 10336645 | 2180.92 | 1850 | 2650 | April 1 | 20.09 | 118.27 | 0.71 | General Creek |
| 16 | 10336660 | 2188.08 | 1850 | 2650 | April 1 | 32.46 | 83.46 | 0.79 | Blackwood Creek |
| 18 | 11266500 | 2576.54 | 1150 | 3950 | April 1 | 836.15 | 140.18 | 0.67 | Merced River |


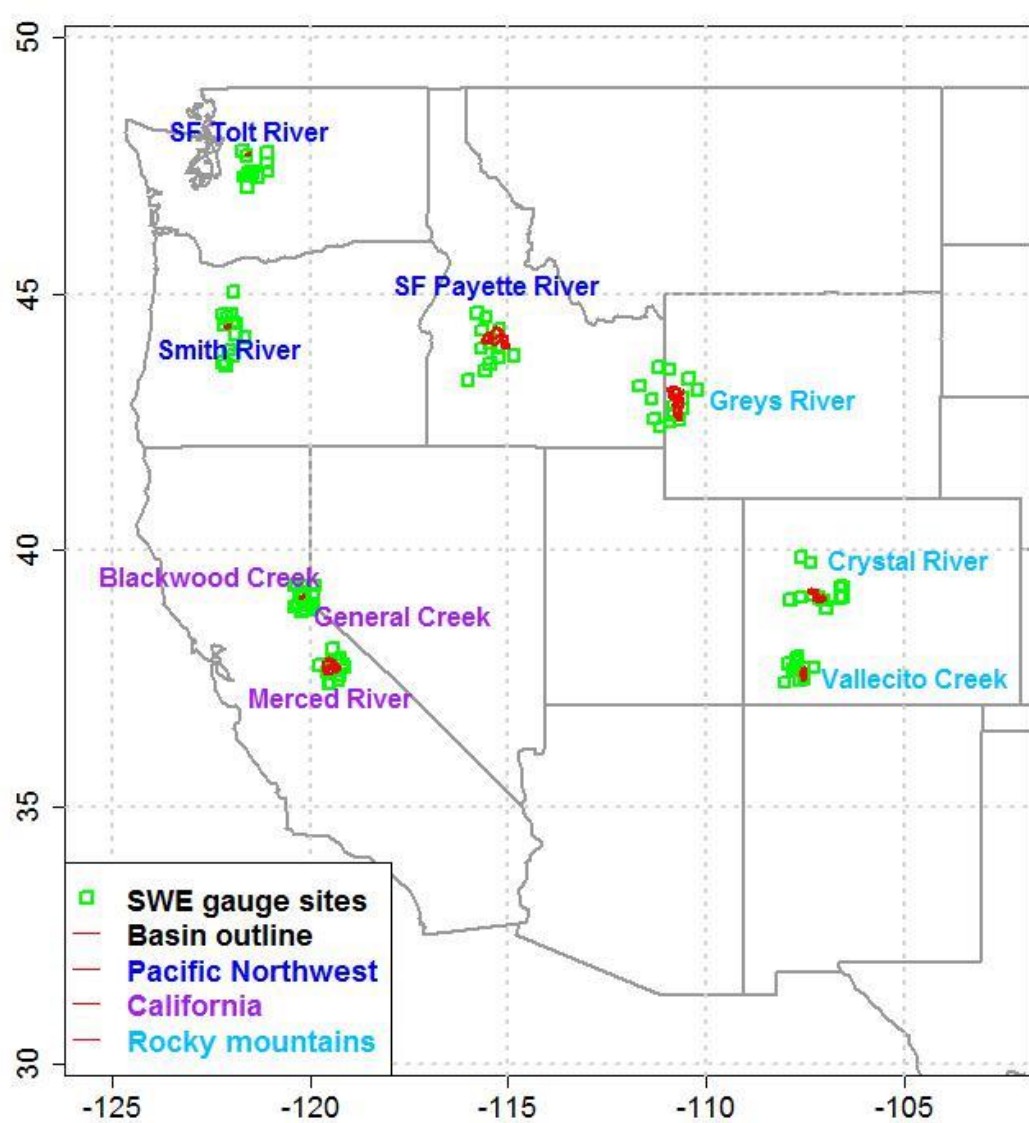


Figure 1. Location of nine case basins in the Western United States (US) and Snow Water Equivalent (SWE) gauge sites.

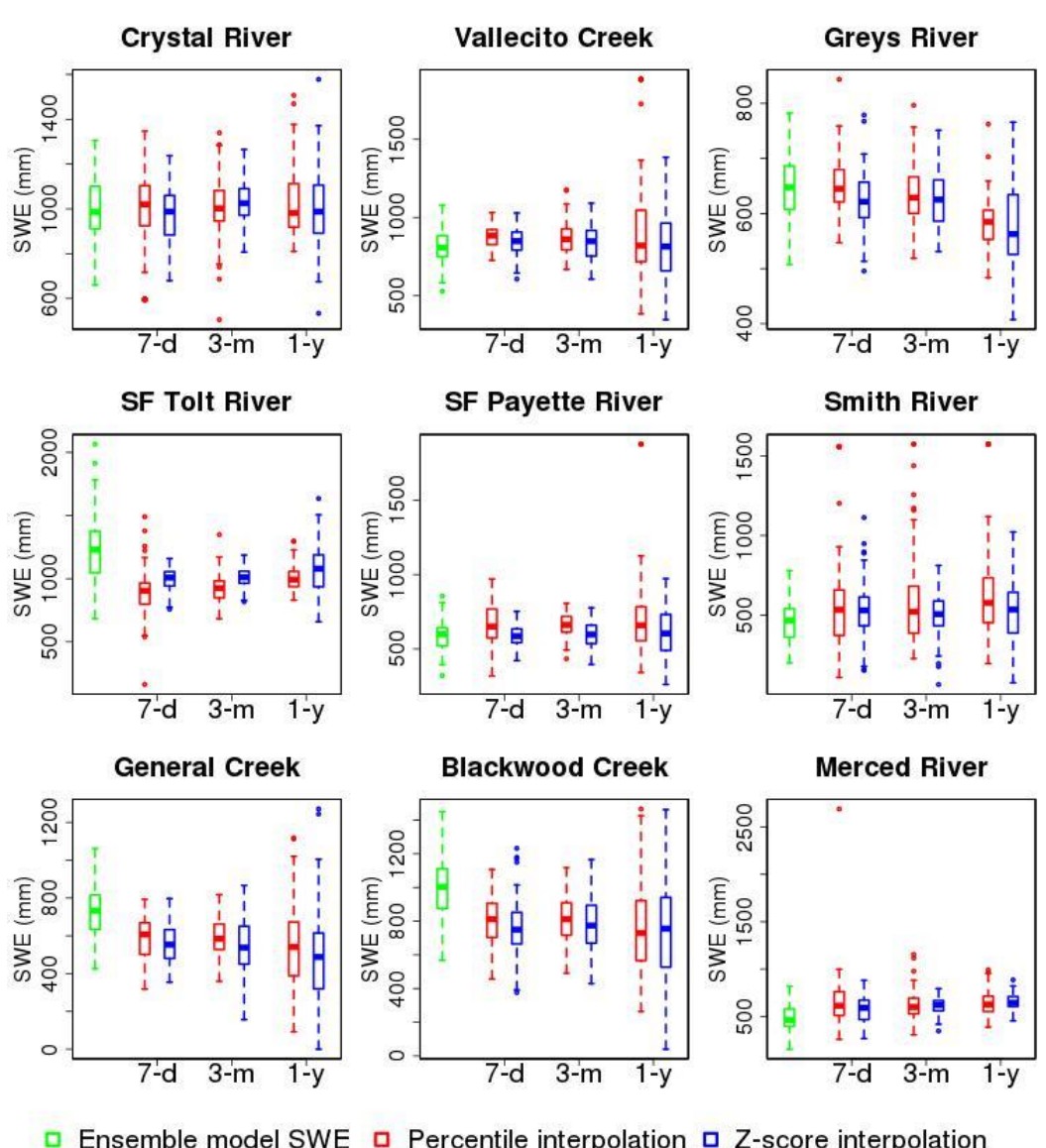

**Figure 2. Boxplots of ensemble model SWE and estimated ensemble SWE observations for the nine case basins on the data assimilation date in 2004, for three window lengths – 7 days, 3 months, and 1 year.**

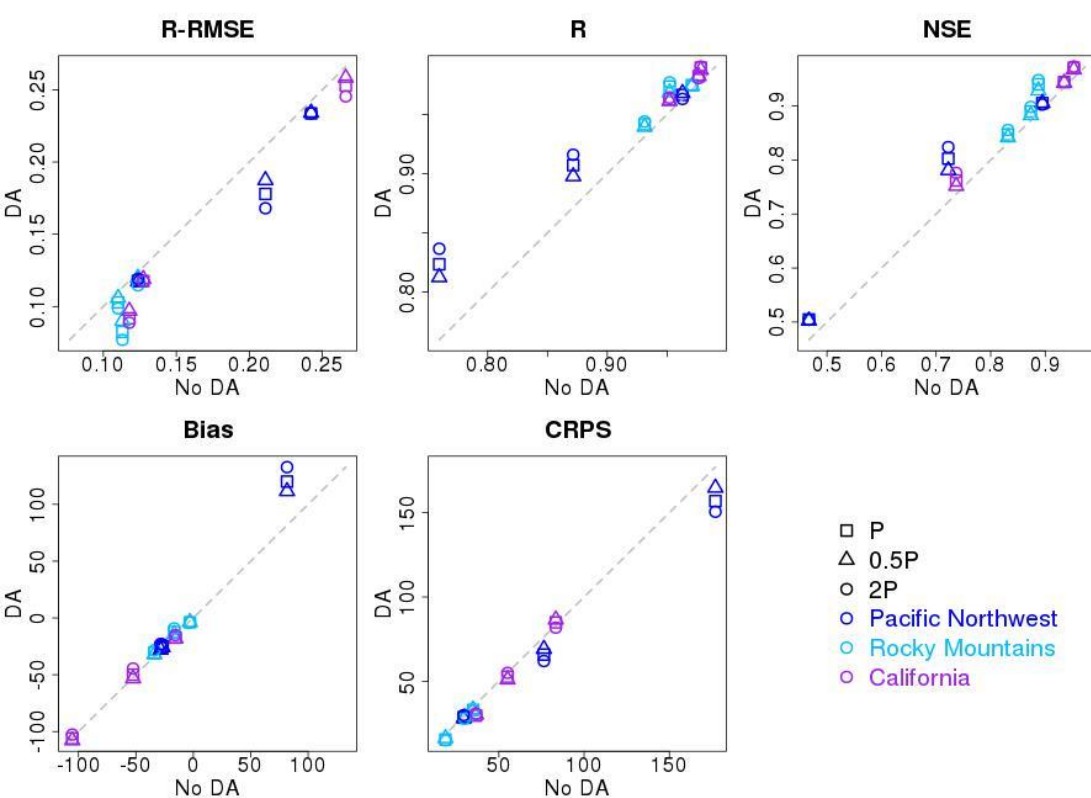

515

**Figure 3. Evaluation metrics for April-July ensemble mean streamflow from the percentile-based interpolation method for the nine case basins using perfect forcing. The verification metrics from upper left to lower right are: R-RMSE is the relative (normalized) root mean squared error, R is the linear (Pearson) correlation coefficient, NSE is the Nash-Sutcliffe Efficiency, bias is the same as mean error, and CRPS is the continuous ranked probability skill scores.**




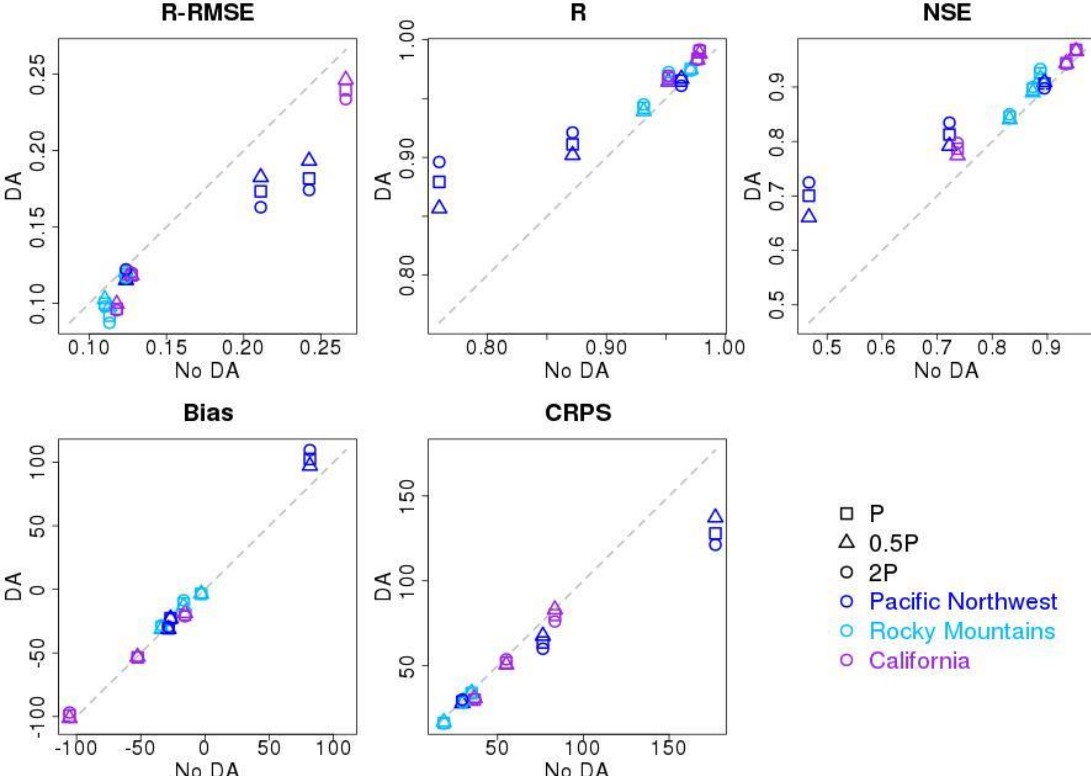


**Figure 4. Evaluation metrics for April-July ensemble mean streamflow from the *Z*-score interpolation for the nine case basins using perfect forcing. The verification metrics from upper left to lower right are: R-RMSE is the relative (normalized) root mean squared error, R is the linear (Pearson) correlation coefficient, NSE is the Nash-Sutcliffe Efficiency, bias is the same as mean error, and CRPS is the continuous ranked probability skill scores.**

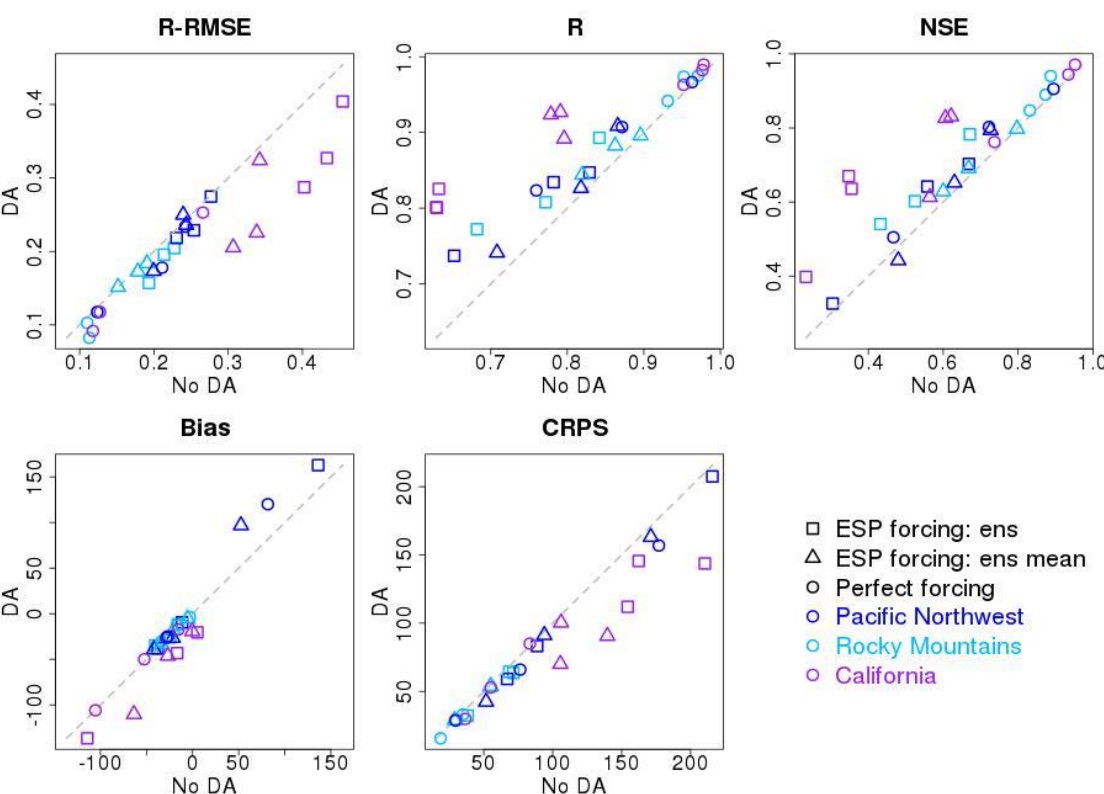

528

**Figure 5. Evaluation statistics of percentile interpolation for the nine case basins with the two variations of Ensemble Streamflow Prediction (ESP) and with perfect forcing data (ens in the legend denotes ensemble). The verification metrics from upper left to lower right are: R-RMSE is the relative (normalized) root mean squared error, R is the linear (Pearson) correlation coefficient, NSE is the Nash-Sutcliffe Efficiency, bias is the same as mean error, and CRPS is the continuous ranked probability skill scores.**

534

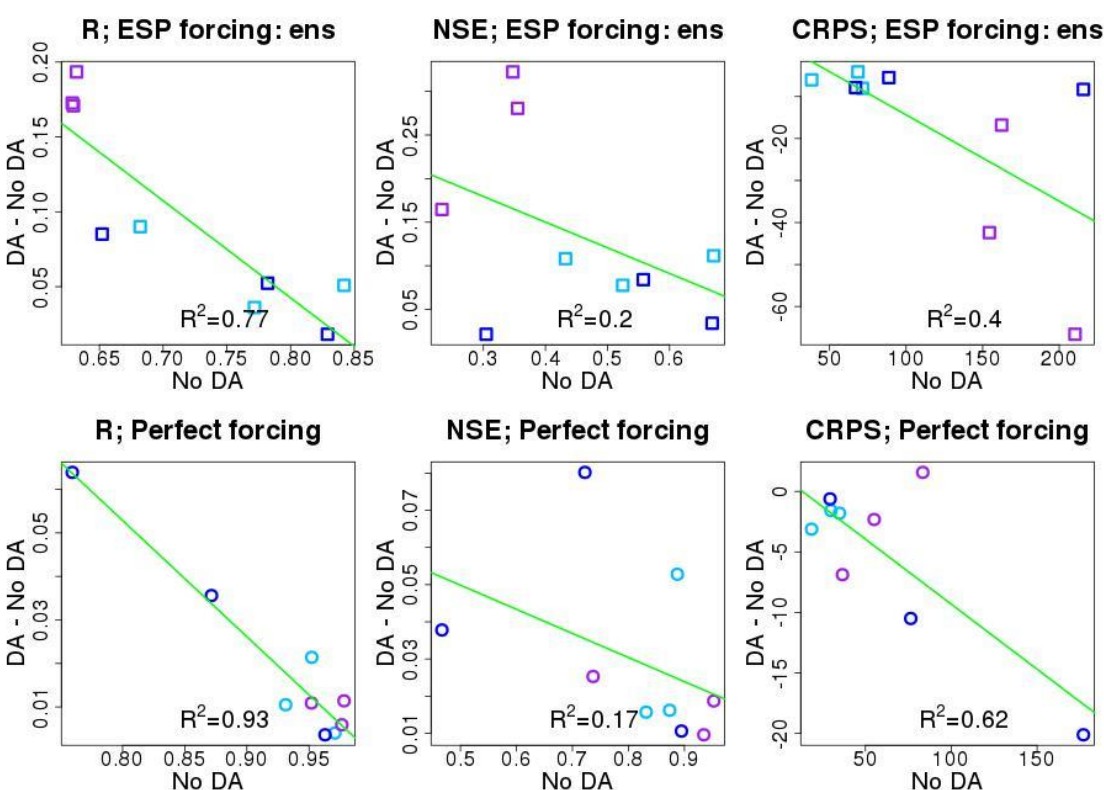

Figure 6. Incremental change in evaluation statistics for Ensemble Streamflow Prediction (ESP) and perfect forcing forecasts using percentile-based interpolation for the nine case basins. R is the linear (Pearson) correlation coefficient, NSE is the Nash-Sutcliffe Efficiency, and CRPS is the continuous ranked probability skill score.

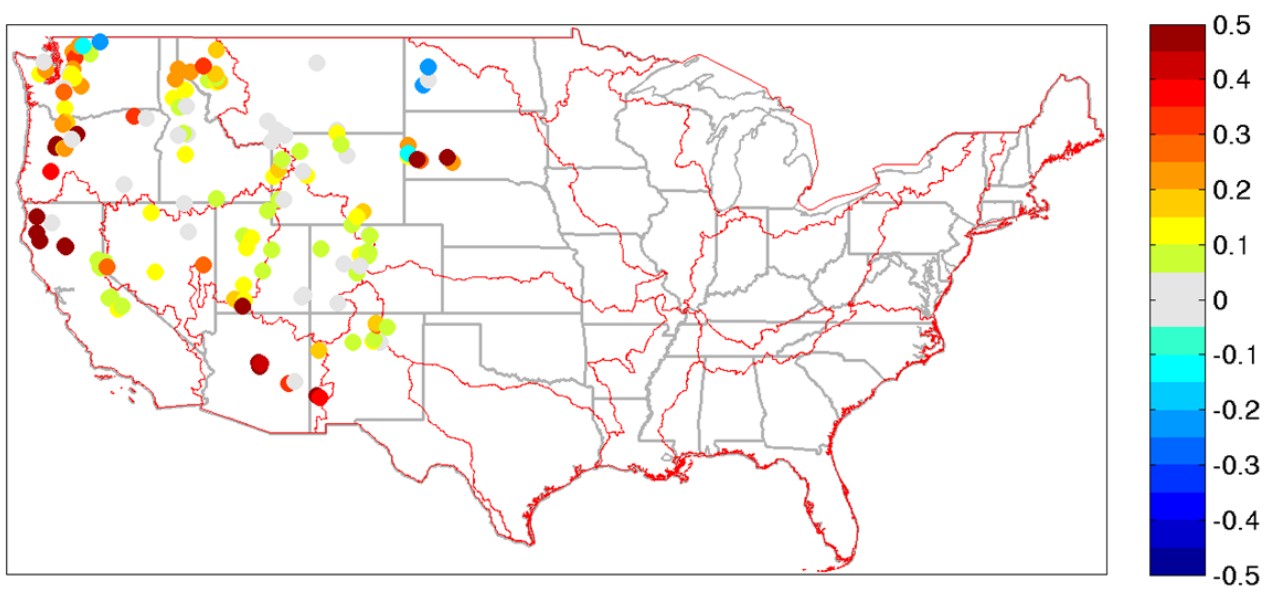


**Figure 7. Difference of the rank correlation of SWE and runoff from the best SNOTEL site (of nearest 10) and**

**calibrated model without DA.**



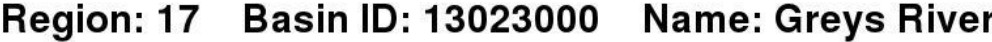


**Figure 8. Time series plots for runoff and SWE for Greys River for water year 1997. Light blue lines indicate individual**


**ensemble member traces. Vertical black dashed line denotes the data assimilation (DA) date.**



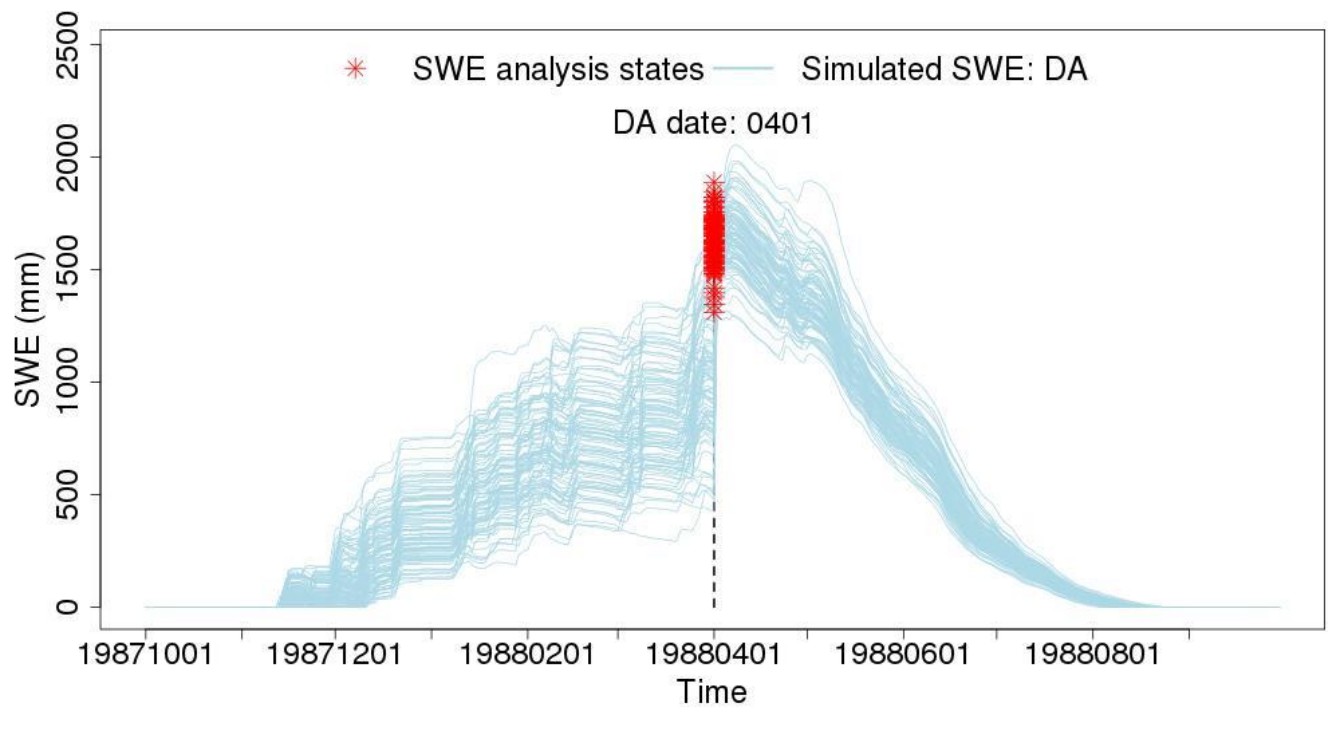

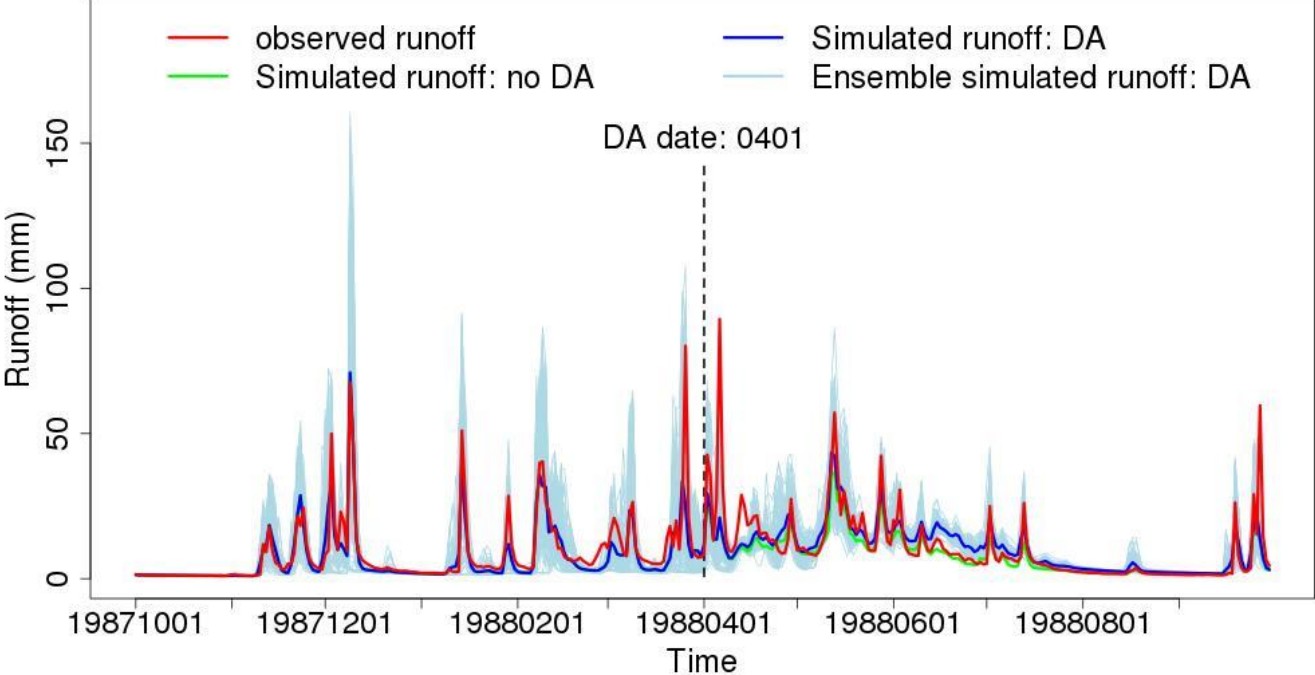


**Figure 9. Time series plots for runoff and SWE for the South Fork (SF) of the Tolt River for water year 1988. Light**

**blue lines indicate individual ensemble member traces. Vertical black dashed line denotes the data assimilation (DA)**

**date.**


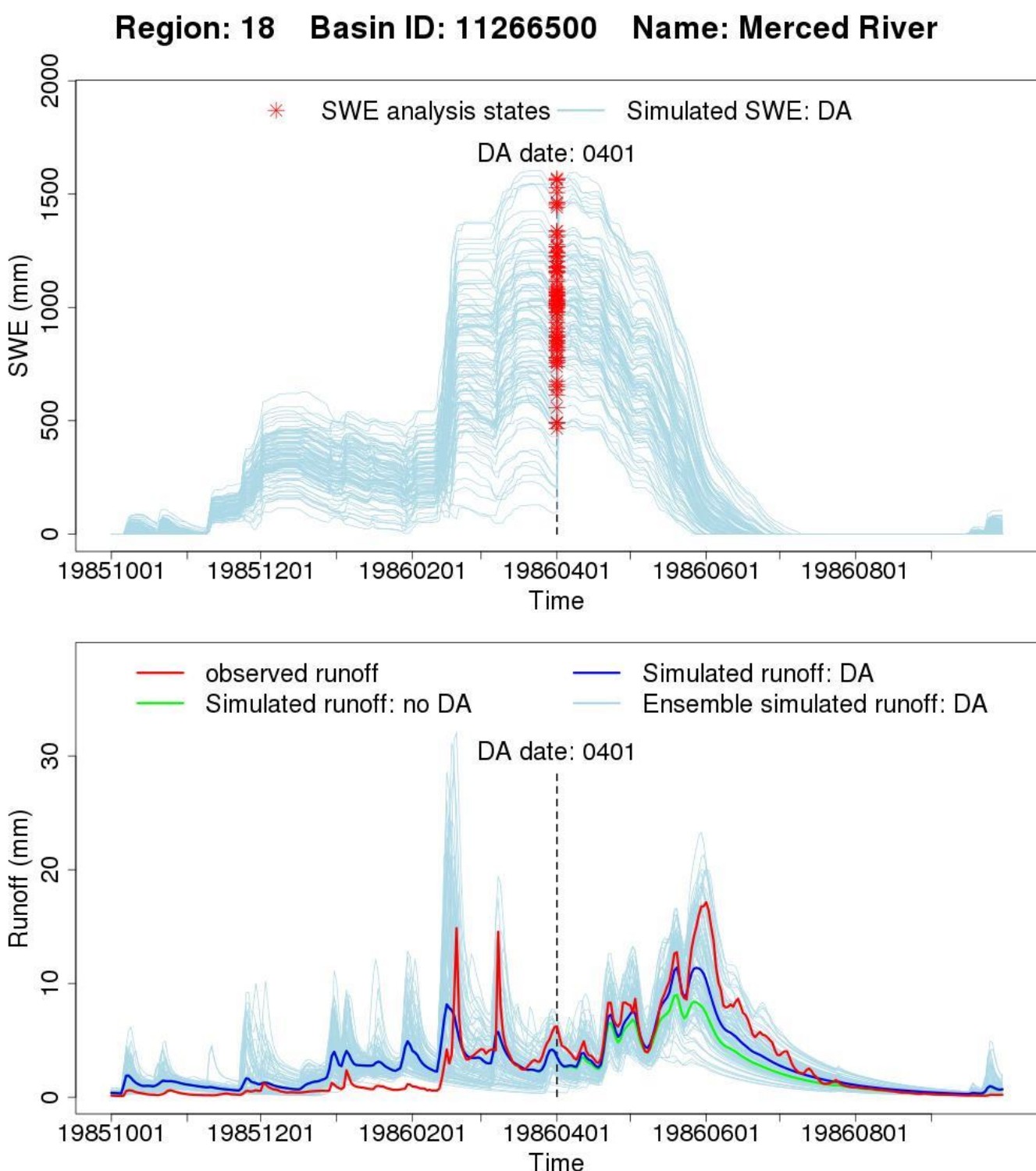

**Figure 10. Time series plots for runoff and SWE for the Merced River for water year 1986. Light blue lines indicate individual ensemble member traces. Vertical black dashed line denotes the data assimilation (DA) date.**

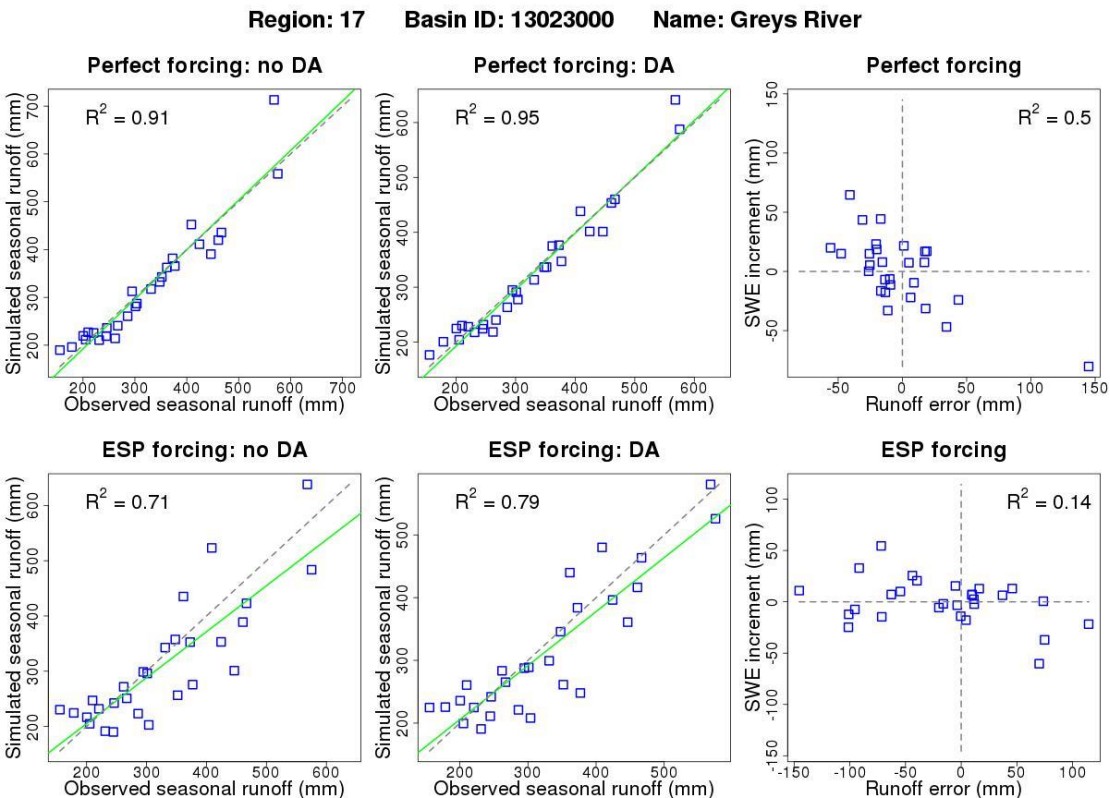

561 **Figure 11. Scatter plots for seasonal runoff and SWE on the data assimilation (DA) date for the Greys River. Black**

562 **dashed diagonal lines are the 1:1 line, while the green lines indicates linear regression fits to data. Perfect forcing results**

563 **are shown in the top row, while Ensemble Streamflow Prediction (ESP) results are in the bottom row.**

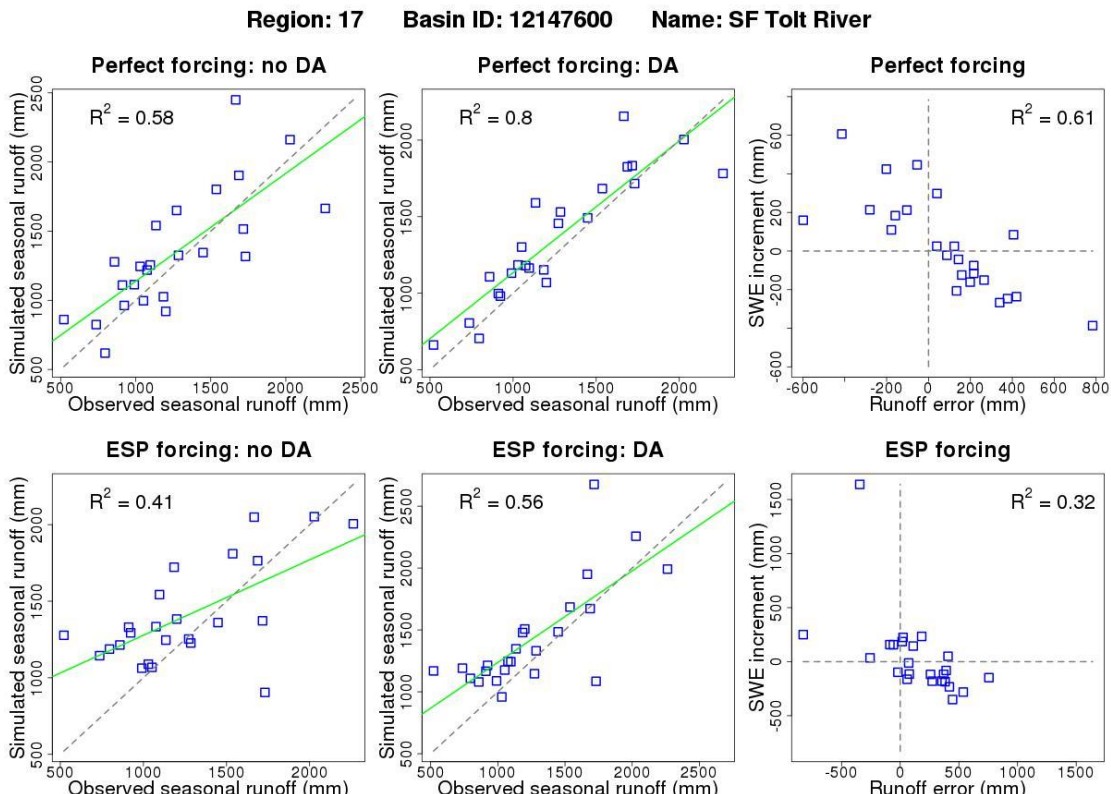

**Figure 12. Scatter plots for seasonal runoff and SWE on the data assimilation (DA) date for the South Fork of the Tolt River. Black dashed diagonal lines are the 1:1 line, while the green lines indicates linear regression fits to data. Perfect forcing results are shown in the top row, while Ensemble Streamflow Prediction (ESP) results are in the bottom row.**


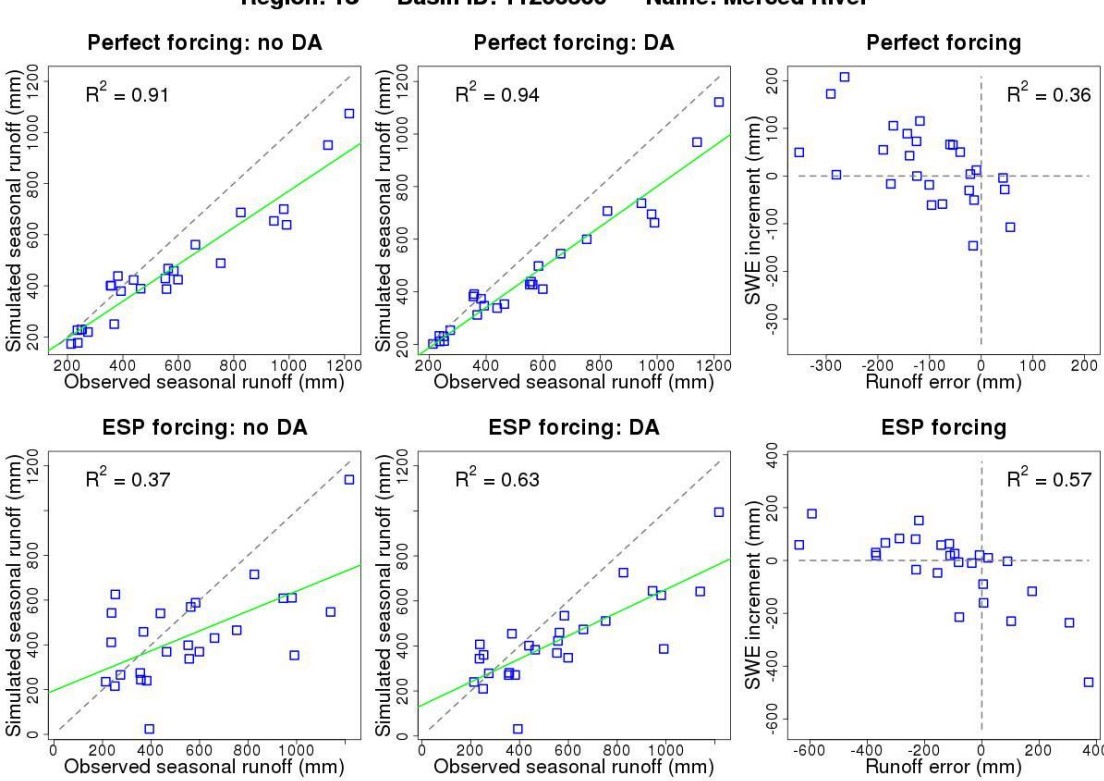


Figure 13. Scatter plots for seasonal runoff and SWE on data assimilation date (DA) for Merced River. Black dashed

diagonal lines are the 1:1 line, while the green lines indicates linear regression fits to data. Perfect forcing results are

shown in the top row, while Ensemble Streamflow Prediction (ESP) results are in the bottom row.


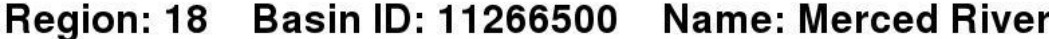

Figure 14. Time series plots for runoff and SWE for the Merced River for water year 1984. Light blue lines indicate individual ensemble member traces. Vertical black dashed line denotes the data assimilation (DA) date.