# Peer review of "Evaluation of snow data assimilation using the Ensemble Kalman Filter for seasonal streamflow prediction in the Western United States"

_Hydrology and Earth System Sciences, 2016_

## Referee Comment (RC1) · K. Engeland (Referee) · 20 Jun 2016

**General comments**

The paper is interesting and deserves publication after a moderate revision. The scientific content and the modelling experiment carried out is excellent. I think, however, that the presentation and discussion can be improved in several ways.

**Structure of paper**

I think there are at least two ways to improve the structure of the paper 1. It could be helpful if you in the introduction provide some explicit aims, objectives, hypotheses or research questions you want two answer, and provide the answers to those in the

conclusions. 2. You discuss your results to a large degree in the results-chapter as well as in this discussion and conclusion chapter. It might be better to make a Results and discussion-chapter and make a much shorter conclusion chapter. Now follows comments to each chapter of the paper.

**1 Introduction**

In the introduction a couple of references could be added: (Griessinger et al., 2016) and (Bergeron et al., 2016) are very fresh paper in this journal and could be included. Some background material from Scandinavian assimilation experiments could be added, see (Udnæs et al., 2007) and (Engeset et al., 2003) for Norway and (Arheimer et al., 2011) for Sweden. For the background information, I think it could be interesting to add a few sentences how data assimilation is used operationally in western US. I guess there are several reports (grey literature) that covers this topic, and that in many cases subjective methods are used. On page 6 lines 148-49 your write a bit about manual practice, this could be moved to the introduction as a background information.

**2 Models and calibration**

I would like to have some more details on the snow model: (1) Do you divide the catchment into elevation zones? This is standard for operational forecasting models in Scandinavia and is important for the performance in catchments with seasonal snow cover. (2) Do you have any sub-catchment distribution of snow (uniform, gamma, log-normal) or do you consider the snow depth to be equal all over the catchment? In Table 1 you list the mean elevation (please specify) but it would also be interesting to show the min and max elevation.

**3.2 Generating ensembles of estimated observed watershed SWE**

I think the use of the term "observation" is confusing since it might refer both to the point observations and the estimated catchment SWE from the regression equations. Especially "estimated observed watershed SWE" is confusing. Maybe it arise from the

Ensemble Kalman filtering setting where the term "observation" is standard terminology. In the text it is not always evident when "observation" refers to the point measurement and when "observation" refers to the observation based catchment SWE. E.g. in line 111 "observation" refers to point observation, whereas for lines 112 and 113 I am confused if you refer to "observation based catchment SWE""or the point measurements. Some suggestion are to write "observation based SWE", "observation based catchment SWE" or "observed catchment SWE". At least you should use a consistent terminology in order to distinguish between the point measurements and the estimated catchment SWE from the point measurements.

**3.2.1 Percentile-regression**

- In lines 121-122 you write: "within a sample of all SWE observations at the same site within a time-window of +/- n days centered on the date of the observation." For me it is not evident if you then use all SWE observations from the year y, from the years preceding y or from all years in your dataset. Maybe the term "date" means "month and day" in this context and not "year, month, day". Please specify.

- Why do you do the regression on the percentile? Does the percentile give you different information than the observed SWE? Please explain why with some sentences.

- Did you have any challenges since p has a lower and upper bound? On line 125, did you need to truncate the simulated p-values to be between 0 and 100?

- The LOO cross validation approach is similar to the Jackknife approach. What is the difference since you do not call it Jackknife?

- Lines 127-129 could be explained better. Do you calculate the percentile p for each ensemble member in order to get 100 pairs of p and SWE from the model?

Both the observation based and the model based ensembles are random, it is not evident how the transformation works. Do you order the samples of p-values?

- Line 129. Why is capital J used?

**3.2.2 Z-score regression**

- Why do you use the term z-score. It might be a bit confusing since the term "score" is often used for model evaluation

- Lines 140-141: "long-term non-zero mean and standard deviation of the full ensemble model SWE within the time-window of +/- n days". Does this mean that you calculate the mean and standard deviation over a sample of 2*n*100 model simulations?

**3.3 EnKF approach and experimental design**

For the data assimilation, it could be useful to (i) write eq. 5 also without the h operator that is actually not used. (ii) describe in two sentences how the analysis works.

**3.6 Verification metrics**

It could be useful to write for which variables the verification metrics is calculated.

**4 Results**

It is not necessary to put tables 2 and 3 in the paper, move them to supplementary material. Figures 3 and 4 are sufficient. From the text and the figures it is confusing for which variable the evaluation statistics is calculated: On line 216 it is written: "The evaluation statistics for ensemble SWE observations". Whereas in the Figure captions it is written that the evaluation is for ensemble mean streamflow. It is not evident for which period the verification metrics is calculated. In Figure 3 and 4 it is not evident which forcing you use. Is it "perfect forecast" or one of the two ESP forecasts? What

is the difference between the evaluations in Figures 3 and 4 versus Figures 5 and 6. Both pairs of figures show evaluation statistics for streamflow forecasts, but I am not able, based on the text, to tell the difference between the two set of plots. There are two results and comments that seem to be contradicting: Line 229: Comment to Figures 2 and 3: "although the DA does not help correct forecast biases." Line 243-245: Comments to Figures 7,8 and 9: "Increasing the ensemble model SWE through DA will lead to increased model runoff, and vice versa. For basins with a strong seasonal cycle of streamflow (e.g. Greys and Merced River), SWE DA generally improves daily runoff forecasts in addition to seasonal volume forecast improvements" How is it possible that DA does not help correct forecast biases whereas it improves seasonal volume forecasts?

**5 Discussion and conclusion:**

- In general, it is helpful if you refer to specific tables and figures in the discussion to make it evident which results you discuss.

- Lines 264-273 could be moved to section 3.2 since it is a good description of the method used and not a discussion of the results presented in this paper.

- I would like more discussion of Figures 10-12, and I would like to know how often the DA improves the simulated seasonal runoff and how often it becomes worse. Figures 7-9 could also include on year when DA makes the simulated seasonal runoff worse. For the subplots to the right in Figures 10-12, it could be interesting to know more about the cases when the points are located in the lower left or upper right quadrants, i.e. to little/much runoff is simulated and you decrease/increase the simulated runoff.

**Details:**

(i) Why is ESP an abbrevation for "Ensemble Streamflow Forecast"? (ii) What is X in 1981-201X on line 174?

**Suggested references:**

Arheimer, B., Lindström, G., Olsson, J., 2011. A systematic review of sensitivities in the Swedish flood-forecasting system. Atmos. Res. 100, 275–284. doi:10.1016/j.atmosres.2010.09.013

Bergeron, J.M., Trudel, M., Leconte, R., 2016. Combined assimilation of streamflow and snow water equivalent for mid-term ensemble streamflow forecasts in snow-dominated regions. Hydrol. Earth Syst. Sci. Discuss. 1–34. doi:10.5194/hess-2016-166

Engeset, R.V., Udnæs, H.C., Guneriussen, T., Koren, H., Malnes, E., Solberg, R., Alfnes, E., 2003. Improving runoff simulations using satellite-observed time-series of snow covered area. Nord. Hydrol. 34, 281–294.

Griessinger, N., Seibert, J., Magnusson, J., Jonas, T., 2016. Assessing the benefit of snow data assimilation for runoff modelling in alpine catchments. Hydrol. Earth Syst. Sci. Discuss. 1–18. doi:10.5194/hess-2016-37

Udnæs, H.-C., Alfnes, E., Andreassen, L.M., 2007. Improving runoff modelling using satellite-derived snow covered area? Nord. Hydrol. 38, 21. doi:10.2166/nh.2007.032

---

## Referee Comment (RC2) · Anonymous Referee #2 · 4 Sep 2016

I found the topic relevant to HESS and a contribution to DA understanding for water resources in snow-dominated watersheds. While I found the paper well written, it was often difficult to follow because of the number of DA-model scenarios and corresponding acronyms (though I struggled to come up with good alternatives). I also thought the results section lacked specifics and overly asked the reader to interpret the figures/tables. Finally, I found the major contribution of the paper to be its potential utility for improving streamflow prediction in watersheds with relatively low model skill. I would like to see the authors leverage their previous work to highlight the utility of the approach presented. It should be noted that I reviewed 'version 2' of the manuscript.

Comment 1: Include more detail in the results. The reader is left to do most of the work

in interpreting and quantifying many statements. Tell us how much and where things were improved and where they were not. Statements like this on line 211: "However, we also note that the ensemble observations of 7-day window can have a large variance, likely due to the more limited sample size for the regression, which can negatively impact DA performance (see Supplement Tables S1.1 and S1.2)." would strongly benefit from specific number. What is large variance? What is a negative impact to DA? I became frustrated having to look at all the figures and table to understand what was meant by sentences like this. A number of examples are listed below, but I encourage the authors to re-read the manuscript to address this problem completely. Lines 221-225: Where by how much? Line 227-228: Which basins? By how much? Lines 243-246: Improves runoff forecasts by how much

Comment 2: Can you remove some of the acronyms or more clearly explain every acronym in the figure captions.

Comment 3: There should be more discussion of why the DA could make predictions worse and where that occurred. Should we be worried about this for future DA efforts? How might we screen sites to ensure that DA does not make predictions worse?

Comment 4: It seems that one of the major contributions of the paper is pointing out that DA methods are likely only make improvements in snow dominated watersheds when model performance was <0.80 NSE. Given that Newman et al., 2015a has quantified the performance of SAC-SMA skill in >500 watersheds, I think a major contribution would be to discuss how many watersheds could benefit from DA and how they are spatially distributed. I think that this should be discussed in the context of where the DA methods did not perform well, i.e. comment 2.

Minor comments: 1. It seems odd to combine the discussion and conclusions section.

---

## Editor Comment (EC1) · I.G. Pechlivanidis (Editor) · 5 Sep 2016

Dear authors,

Your manuscript was reviewed by two experts on seasonal hydro-meteorological forecasting. Overall, the reviewers responded positively on the importance of this investigation. However, some major issues still remain. I found their comments crucial and addressing them would improve the quality of your article. I encourage you to submit your response to allow an open discussion and also a preliminary evaluation of the potentially revised manuscript, which will eventually result in a publication in this special issue.

[Figure]

I am looking forward to your reply.

Best regards,

Ilias Pechlivanidis

---

## Author Comment (AC1) · 6 Oct 2016

We greatly appreciate referee K. Engeland for your thoughtful and positive comments on this manuscript. Below are our detailed responses to the points raised.
General comments The paper is interesting and deserves publication after a moderate revision. The scientific content and the modelling experiment carried out is excellent. I

think, however, that the presentation and discussion can be improved in several ways.

**Response: Thank you for the positive comment.**

Structure of paper I think there are at least two ways to improve the structure of the paper 1. It could be helpful if you in the introduction provide some explicit aims, objectives, hypotheses or research questions you want two answer, and provide the answers to those in the conclusions. 2. You discuss your results to a large degree in the results-chapter as well as in this discussion and conclusion chapter. It might be better to make a Results and discussion-chapter and make a much shorter conclusion chapter. Now follows comments to each chapter of the paper.

**Response to comment 1 on the structure of the paper: We agree with this point. It would be good to add more explicit motivation and specific questions examined in the paper to the introduction. Corresponding answers would be added to the conclusions as you note.**

**Response to comment 2 on the structure of the paper: We agree with this point as well. It is possible to include a vast majority of the discussion in a new Results and Discussion section. We'd likely then rename the final section to "Summary" and include only key discussion points for reference with concluding statements and answers to the study questions.**

1 Introduction In the introduction a couple of references could be added: (Griessinger et al., 2016) and (Bergeron et al., 2016) are very fresh paper in this journal and could be included. Some background material from Scandinavian assimilation experiments could be added, see (Udnæs et al., 2007) and (Engeset et al., 2003) for Norway and (Arheimer et al., 2011) for Sweden. For the background information, I think it could be interesting to add a few sentences how data assimilation is used operationally in western US. I guess there are several reports (grey literature) that covers this topic, and that in many cases subjective methods are used. On page 6 lines 148-49 your write a bit about manual practice, this could be moved to the introduction as a background
information.

**Response: Thank you for the references, they will be great to add to the introduction. As mentioned in lines 148-149, snow data assimilation is implemented manually in operation currently. We plan to move this to the introduction with a few more sentences describing the state of operational DA in the Western US. This helps more clearly defines the motivation of the paper.**

2 Models and calibration I would like to have some more details on the snow model: (1) Do you divide the catchment into elevation zones? This is standard for operational forecasting models in Scandinavia and is important for the performance in catchments with seasonal snow cover. (2) Do you have any sub-catchment distribution of snow (uniform, gamma, lognormal) or do you consider the snow depth to be equal all over the catchment? In Table 1 you list the mean elevation (please specify) but it would also be interesting to show the min and max elevation.

**Response: We did not divide the catchment into elevation zones currently. We agree that elevation bands are standard practice in many regions, including the Western US. For this study, the reference models (no DA simulations) are also lumped, thus we feel the DA work and improvements are still relevant. We are working toward elevation band simulations with DA across many basins currently, but it is not included in this manuscript. The snow model assumes uniform depth across the basin, but does have an empirical snow covered area curve (see Snow-17 references in this paper). We will add the min and max elevation in Table 1.**

3.2 Generating ensembles of estimated observed watershed SWE I think the use of the term "observation" is confusing since it might refer both to the point observations and the estimated catchment SWE from the regression equations. Especially "estimated observed watershed SWE" is confusing. Maybe it arise from the Ensemble Kalman filtering setting where the term "observation" is standard terminology. In the text it is not

always evident when "observation" refers to the point measurement and when "observation" refers to the observation based catchment SWE. E.g. in line 111 "observation" refers to point observation, whereas for lines 112 and 113 I am confused if you refer to "observation based catchment SWE""or the point measurements. Some suggestion are to write "observation based SWE", "observation based catchment SWE" or "observed catchment SWE". At least you should use a consistent terminology in order to distinguish between the point measurements and the estimated catchment SWE from the point measurements.

**Response: We agree with this point and will clarify and change the terminology upon revision.**

3.2.1 Percentile-regression In lines 121-122 you write: "within a sample of all SWE observations at the same site within a time-window of +/- n days centered on the date of the observation." For me it is not evident if you then use all SWE observations from the year y, from the years preceding y or from all years in your dataset. Maybe the term "date" means "month and day" in this context and not "year, month, day". Please specify.

**Response: We mean all the years in our dataset, and will revise the sentences in our manuscript accordingly.**

Why do you do the regression on the percentile? Does the percentile give you different information than the observed SWE? Please explain why with some sentences.

**Response: We did this mainly for reducing interpolation uncertainty caused by spatial heterogeneity of SWE gauge sites following Slater and Clark (2006). We will include clarification in the revised draft.**

Did you have any challenges since p has a lower and upper bound? On line 125, did you need to truncate the simulated p-values to be between 0 and 100?

**Response: Yes, we needed to truncate the simulated p-values to between 0 and**

**100. We experimented with various assumptions related to this truncation and found that straight truncation (e.g. a regression percentile of 110 is set to 100) worked the best in these cases.**

The LOO cross validation approach is similar to the Jackknife approach. What is the difference since you do not call it Jackknife.

**Response: We call it LOO cross validation approach just for specifying that it is a special Jackknife approach that only one sample is cut out each time.**

Lines 127-129 could be explained better. Do you calculate the percentile p for each ensemble member in order to get 100 pairs of p and SWE from the model? Both the observation based and the model based ensembles are random, it is not evident how the transformation works. Do you order the samples of p-values?

**Response: Yes, we calculated the percentile p for each ensemble member in order to get 100 pairs of p and SWE from the model. We will include the following discussion to improve clarity. We did not order the samples of p-values. We just calculate the corresponding percentiles of the full ensemble model SWE (i.e., a sample of (2n+1)*Y*100 members, where (2n+1) is the length of time window each year, Y is the total number of years of our dataset, 100 is the number of ensemble model SWE members) according to the estimated sample p-values.**

Line 129. Why is capital J used?

**Response: Initially we used capital J just for distinguishing it from the index of percentile ( ), but the capital J does seem unnecessary. We will revise to use just lowercase j in our manuscript.**

3.2.2 Z-score regression Why do you use the term z-score. It might be a bit confusing since the term "score" is often used for model evaluation

**Response: It is just a conventional name referring to the transformation of Eq. (1). When using a Gaussian distribution, distance from the mean is often**

**discussed in terms of standard deviations, and when normalized by the standard deviation of a particular distribution, a deviation is termed "Z-score" for Z-standard deviations from the mean in statistics.**

Lines 140-141: "long-term non-zero mean and standard deviation of the full ensemble model SWE within the time-window of +/- n days". Does this mean that you calculate the mean and standard deviation over a sample of 2*n*100 model simulations?

**Response: We calculated the mean and standard deviation over a sample no greater than (2n+1)*Y*100 (only non-zero members are used), where (2n+1) is the length of time window of +/- n days in each year, Y is the total number of years of our datasets, and 100 is the number of ensemble model SWE members.**

3.3 EnKF approach and experimental design For the data assimilation, it could be useful to (i) write eq. 5 also without the h operator that is actually not used. (ii) describe in two sentences how the analysis works.

**Response: We prefer to keep the transformation vector h as it is the formal terminology and if the transformation is not done in a pre-processing step as we have done it does need to be performed. We calculate an analysis via eq. 5 and use that analysis to update the Snow-17 SWE states. We then run the model system with the updated states until the end of the WY. This clarification would be included as revised text at the end of section 3.3.**

3.6 Verification metrics It could be useful to write for which variables the verification metrics is calculated.

**Response: The verification metrics are for seasonal streamflow volume. Text will be modified to state this.**

4 Results It is not necessary to put tables 2 and 3 in the paper, move them to supplementary material. Figures 3 and 4 are sufficient.

**Response: We will move them to the supplementary material upon revision.**

From the text and the figures it is confusing for which variable the evaluation statistics is calculated: On line 216 it is written: "The evaluation statistics for ensemble SWE observations". Whereas in the Figure captions it is written that the evaluation is for ensemble mean streamflow. It is not evident for which period the verification metrics is calculated.

**Response: Our evaluation statistics are all calculated for streamflow. We can clarify the text to clearly state that the evaluation metrics are for seasonal streamflow volume, applied to the two SWE interpolation approaches using varying window lengths for the SWE transformation.**

In Figure 3 and 4 it is not evident which forcing you use. Is it "perfect forecast" or one of the two ESP forecasts? What is the difference between the evaluations in Figures 3 and 4 versus Figures 5 and 6. Both pairs of figures show evaluation statistics for streamflow forecasts, but I am not able, based on the text, to tell the difference between the two set of plots.

**Response: In figures 3 and 4, perfect forcing is used. This will be clarified in the text and figure captions. The difference between the evaluations in Figures 3 and 4 versus Figures 5 and 6 is that their focuses are different. Figure 3 and 4 show the evaluation results of the sensitivity analysis of model and observation error variance (i.e., P, 0.5P and 2P). Figure 5 and 6 show the evaluation results of seasonal ESP (two types of ESP forecasts) compared with perfect forcing. Further clarification will be added to the text.**

There are two results and comments that seem to be contradicting: Line 229: Comment to Figures 2 and 3: "although the DA does not help correct forecast biases." Line 243-245: Comments to Figures 7,8 and 9: "Increasing the ensemble model SWE through DA will lead to increased model runoff, and vice versa. For basins with a strong seasonal cycle of streamflow (e.g. Greys and Merced River), SWE DA generally improves daily runoff forecasts in addition to seasonal volume forecast improvements"

How is it possible that DA does not help correct forecast biases whereas it improves seasonal volume forecasts?

**Response: We believe this comes through modification of both negative and positive runoff errors. Bias is a sum of signed errors, thus the noDA and DA runs can have similar total error even if the no DA run has large year to year errors. The DA run can improve a statistic like RMSE which is a squared error metric without changing bias. For example Figure 12, lower left panels highlights that DA reduces large positive error for a few years and conversely, increases negative runoff error over many years. This improves correlation, RMSE, etc, but leave bias nearly unchanged.**

5 Discussion and conclusion: In general, it is helpful if you refer to specific tables and figures in the discussion to make it evident which results you discuss.

**Response: We will revise the paper to include more specifics figure and table references in the summary section.**

Lines 264-273 could be moved to section 3.2 since it is a good description of the method used and not a discussion of the results presented in this paper.

**Response: Agreed, this will be moved.**

I would like more discussion of Figures 10-12, and I would like to know how often the DA improves the simulated seasonal runoff and how often it becomes worse. Figures 7-9 could also include on year when DA makes the simulated seasonal runoff worse. For the subplots to the right in Figures 10-12, it could be interesting to know more about the cases when the points are located in the lower left or upper right quadrants, i.e. to little/much runoff is simulated and you decrease/increase the simulated runoff.

**Response: Reviewer 2 had a similar comment to this. Most of this response is repeated there as well. Generally, when the SWE increment is incorrect, it is less than 10The Merced River is the only basin to use state of California SWE ob-**

servations, and these may be of lower quality as evidenced by the large amount of manual quality control we had to perform on the data and the quality control discussion of these data in Lundquist et al. (2015). This suggests that observed SWE data need to be of higher quality (or information content) than the calibrated model SWE to have a positive impact in the DA system. Conversely, there are years where the noDA runoff error is large, but the SWE increment is small in all three basins. This is not unexpected as spring SWE is not perfectly correlated with subsequent runoff. This may also hint at a level of data loss in the EnKF and modeling system, future work should compare streamflow hindcasts using this type of system with traditional statistical methods using SWE as a primary input.

We will look a bit more into these years and try to identify if there was anything that makes them "special."

Reference: Lundquist, J. D., M. Hughes, B. Henn, E. D. Gutmann, B. Livneh, J. Dozier, and P. Neiman, 2015: High-elevation precipitation patterns: using snow measurements to assess daily gridded datasets across the Sierra Nevada, California. J. Hydrometeorology, 16, 1773-1792. doi: 10.1175/JHM-D-15-0019.1.

Details: (i) Why is ESP an abbrevation for "Ensemble Streamflow Forecast"? (ii) What is X in 1981-201X on line 174?

Response: We will fix the text, ESP should be an abbreviation for "Ensemble Streamflow Prediction".

Suggested references: Arheimer, B., Lindström, G., Olsson, J., 2011. A systematic review of sensitivities in the Swedish flood-forecasting system. Atmos. Res. 100, 275–284. doi:10.1016/j.atmosres.2010.09.013 Bergeron, J.M., Trudel, M., Leconte, R., 2016. Combined assimilation of streamflow and snow water equivalent for mid-term ensemble streamflow forecasts in snowdominated regions. Hydrol. Earth Syst. Sci. Discuss. 1–34. doi:10.5194/hess-2016-166 Engeset, R.V., Udnæs, H.C., Guneriussen,

T., Koren, H., Malnes, E., Solberg, R., Alfnes, E., 2003. Improving runoff simulations using satellite-observed time-series of snow covered area. Nord. Hydrol. 34, 281–294. Griessinger, N., Seibert, J., Magnusson, J., Jonas, T., 2016. Assessing the benefit of snow data assimilation for runoff modelling in alpine catchments. Hydrol. Earth Syst. Sci. Discuss. 1–18. doi:10.5194/hess-2016-37 Udnæs, H.-C., Alfnes, E., Andreassen, L.M., 2007. Improving runoff modelling using satellite-derived snow covered area? Nord. Hydrol. 38, 21. doi:10.2166/nh.2007.032 Interactive comment on Hydrol. Earth Syst. Sci. Discuss., doi:10.5194/hess-2016-185, 2016.

---

## Author Comment (AC2) · 6 Oct 2016

**We greatly appreciate this anonymous referee for your thoughtful and positive comments on this manuscript. We have revised the manuscript accordingly. Below are detailed responses to the points raised. Our responses are in bold.**

Anonymous Referee 2,

I found the topic relevant to HESS and a contribution to DA understanding for water resources in snow-dominated watersheds. While I found the paper well written, it was often difficult to follow because of the number of DA-model scenarios and corresponding acronyms (though I struggled to come up with good alternatives). I also

thought the results section lacked specifics and overly asked the reader to interpret the figures/tables. Finally, I found the major contribution of the paper to be its potential utility for improving streamflow prediction in watersheds with relatively low model skill. I would like to see the authors leverage their previous work to highlight the utility of the approach presented. It should be noted that I reviewed 'version 2' of the manuscript.

**Response: Thank you for your overall summary comments of the paper. We agree with the general comment that additional specific analysis can be added to the results section. We will clarify the text throughout, with emphasis on the results section. Our replies to your specific comments give more detail to this general response.**

Comment 1: Include more detail in the results. The reader is left to do most of the work in interpreting and quantifying many statements. Tell us how much and where things were improved and where they were not. Statements like this on line 211: "However, we also note that the ensemble observations of 7-day window can have a large variance, likely due to the more limited sample size for the regression, which can negatively impact DA performance (see Supplement Tables S1.1 and S1.2)." would strongly benefit from specific number. What is large variance? What is a negative impact to DA?

**Response: The negative impact is truly a reduction in the positive impact of DA when comparing the 7-day window to the 3-month window. We will include specific numbers and clarify this text.**

I became frustrated having to look at all the figures and table to understand what was meant by sentences like this. A number of examples are listed below, but I encourage the authors to re-read the manuscript to address this problem completely. Lines 221-225: Where by how much?

**Response: We agree that the results section needs more analysis clearly stated in the text. We will tabulate key results for the metrics across example metrics for the entire basin set and add those results to the text.**

Line 227-228: Which basins? By how much?

**Response: Again, we will revise this discussion.**

Lines 243-246: Improves runoff forecasts by how much

**Response: We will quantify the improvement to daily flow for the example basins given.**

Comment 2: Can you remove some of the acronyms or more clearly explain every acronym in the figure captions.

**Response: Yes, we agree these need to be more clearly defined for each figure, or removed entirely in the captions. We will do that upon revision.**

Comment 3: There should be more discussion of why the DA could make predictions worse and where that occurred. Should we be worried about this for future DA efforts? How might we screen sites to ensure that DA does not make predictions worse?

**Response: We can see from the right two subplots in Figures 10-12 that the years when DA makes the simulated runoff worse is when runoff error is generally very small. Generally, those SWE increments are less than 10In terms of observational sites, the Merced River is the only basin to use state of California SWE observations, and these may be of lower quality as evidenced by the large amount of manual quality control we had to perform on the data and the quality control discussion of these data in Lundquist et al. (2015). This suggests that observed SWE data need to be of higher quality (or information content) than the calibrated model SWE to have a positive impact in the DA system. Conversely, there are years where the noDA runoff error is large, but the SWE increment is small in all three basins. This is not unexpected as spring SWE is not perfectly correlated with subsequent runoff. This may also hint at a level of data loss in the EnKF and modeling system, future work should compare streamflow hindcasts using this type of system with traditional statistical methods using SWE**

[Figure]

as a primary input. We believe screening of observational sites is a difficult task. The above discussion and our results in California does suggest screening is needed. High quality sites with no information content would also need to be screened as well (also see discussion in reply to comment 4). It is possible that guidelines for this could be developed and then potentially automated, but this is likely a major undertaking. In this study, site selection was first taken using closest distances to the basin, then manual screening of suspect sites and sites that had little relationship with runoff were removed. It is possible some formalization of this methodology could be developed. That being said, the relationship between SWE and runoff will likely be basin dependent and the addition of an assimilation system and model forecast introduces information losses that are also likely basin dependent since the hydrologic modeling system is basin dependent, such that a screening methodology based solely on observations is likely to misidentify potential degradation or improvement when DA is applied.

Comment 4: It seems that one of the major contributions of the paper is pointing out that DA methods are likely only make improvements in snow dominated watersheds when model performance was <0.80 NSE. Given that Newman et al., 2015a has quantified the performance of SAC-SMA skill in >500 watersheds, I think a major contribution would be to discuss how many watersheds could benefit from DA and how they are spatially distributed. I think that this should be discussed in the context of where the DA methods did not perform well, i.e. comment 2.

Response: This idea you mention is an interesting topic. We will look back through the database and add some additional analysis examining spatial location of basins that may benefit from DA using the basic metrics of noDA NSE and contribution of SWE to runoff. That being said, a comprehensive description and analysis about how many watersheds could benefit from DA and how they are spatially distributed is a large topic and could be a separate paper. Preliminary screening of candidate basins would not only require the basic metrics of

[Figure]

**being snow dominated, generally lower noDA skill, but also somehow assessing the quality of information from the nearby observation sites. Furthermore, we'd expect that implementation of the enKF DA would result in potential differences as there may be data loss in the observation transformation operator, etc.**

Minor comments: 1. It seems odd to combine the discussion and conclusions section.

**Response: We will revise the last two sections to be Results and Discussion and Summary. More discussion will be included in section 4, while the summary section will restate key discussion points and then findings of the study.**

**References: Lundquist, J. D., M. Hughes, B. Henn, E. D. Gutmann, B. Livneh, J. Dozier, and P. Neiman, 2015: High-elevation precipitation patterns: using snow measurements to assess daily gridded datasets across the Sierra Nevada, California. J. Hydrometeorology, 16, 1773-1792. doi: 10.1175/JHM-D-15-0019.1.**

---

## Author Comment (AC3) · 22 Nov 2016

Dear Dr. Pechlivanidis,

Thank you for your decision comments. We believe we have addressed yours and the reviewer comments in the uploaded revised manuscript in a manner consistent with our reviewer replies. We are willing to make further changes as you see fit.

Sincerely, Andrew Newman

---

## Author Response (AR2)

Dear Dr. Pechlivanidis,
We thank you for your prompt decision and editorial suggestions.  We have made all of the suggested changes, as well a few
minor updates and corrections to citations and changing the order of several figure panels.  Content is unchanged, merely the
order of the two panels in Figures 8-10, and 14 have been switched.
Sincerely,
Dr. Andrew Newman

[revised manuscript text omitted]

**Region: 18    Basin ID: 11266500    Name: Merced River**

[Figure]

[Figure]

[Figure]

**Region: 18    Basin ID: 11266500    Name: Merced River**

**Figure 14.  Time series plots for runoff and SWE for the Merced River for water year 1984 . Light blue lines indicate individual ensemble member traces.  Vertical black dashed line denotes the data assimilation (DA) date.**

---

## Author Response (AR3)

Dear Dr. Pechlivanidis,

Thank you for the further suggestions to our final changes.  We have edited the figure captions accordingly.

Sincerely,

Andrew Newman

[revised manuscript text omitted]

---

## Author Response (AR4)

Dear Dr. Pechlivanidis,

Thank you for the further suggestions to our final changes.  We have edited the figure captions accordingly.

Sincerely,

Andrew Newman

[revised manuscript text omitted]